# Functional and pharmacological analyses of visual habituation learning in larval zebrafish

**Laurie Anne Lamiré[1], Martin Haesemeyer[2], Florian Engert[3,4†], Michael Granato[5†], Owen Randlett[1*†]**

[1]Laboratoire MeLiS, UCBL - CNRS UMR5284 - Inserm U1314, Institut NeuroMyoGène, Faculté de Médecine et de Pharmacie, Lyon, France; [2]The Ohio State University, Department of Neuroscience, Columbus, United States; [3]Department of Molecular and Cellular Biology, Faculty of Arts and Sciences, Harvard University, Cambridge, United States; [4]Center for Brain Science, Faculty of Arts and Sciences, Harvard University, Cambridge, United States; [5]Department of Cell and Developmental Biology, University of Pennsylvania, Perelman School of Medicine, Philadelphia, United States

**\*For correspondence:**
owen.randlett@univ-lyon1.fr

[†]These authors contributed equally to this work

**Competing interest:** The authors declare that no competing interests exist.

**Abstract** Habituation allows animals to learn to ignore persistent but inconsequential stimuli. Despite being the most basic form of learning, a consensus model on the underlying mechanisms has yet to emerge. To probe relevant mechanisms, we took advantage of a visual habituation paradigm in larval zebrafish, where larvae reduce their reactions to abrupt global dimming (a dark flash). We used $Ca^{2+}$ imaging during repeated dark flashes and identified 12 functional classes of neurons that differ based on their rate of adaptation, stimulus response shape, and anatomical location. While most classes of neurons depressed their responses to repeated stimuli, we identified populations that did not adapt or that potentiated their response. These neurons were distributed across brain areas, consistent with a distributed learning process. Using a small-molecule screening approach, we confirmed that habituation manifests from multiple distinct molecular mechanisms, and we have implicated molecular pathways in habituation, including melatonin, oestrogen, and GABA signalling. However, by combining anatomical analyses and pharmacological manipulations with $Ca^{2+}$ imaging, we failed to identify a simple relationship between pharmacology, altered activity patterns, and habituation behaviour. Collectively, our work indicates that habituation occurs via a complex and distributed plasticity processes that cannot be captured by a simple model. Therefore, untangling the mechanisms of habituation will likely require dedicated approaches aimed at subcomponent mechanisms underlying this multidimensional learning process.

## eLife assessment

This **valuable** manuscript attempts to identify the brain regions and cell types involved in habituation to dark flash stimuli in larval zebrafish. Habituation being a form of learning widespread in the animal kingdom, the investigation of neural mechanisms underlying it is a worthwhile endeavor. The authors use a combination of behavioral analysis, neural activity imaging, and pharmacological manipulation to investigate brain-wide mechanisms of habituation. While the data presented are **solid**, the authors conclude that there is no simple relationship between pharmacological intervention, neural activity patterns, and behavioral outcomes, and a robust causative link can therefore not be established.

## Introduction

A central function of the brain is to learn and change with experience. These adaptations can reflect attempts to identify and attend preferentially to salient stimuli. For example, identifying the smell of a predator or prey may be crucial, while identifying that my home still smells like my kin is not. This ability to suppress responses to continuous non-salient stimuli is known as habituation, a process generally considered to be the simplest form of learning and memory (*Rankin et al., 2009*). Habituation is conserved across all animals, and like other forms of plasticity, exists in at least two mechanistically distinct forms: transient short-term habituation and protein-synthesis-dependent long-term habituation. Here we focus on long-term habituation, which serves as a pragmatic model to dissect plasticity processes in neural circuits.

Work on long-term habituation in various species and paradigms has led to significant insights into the adaptations underlying this process (*Cooke and Ramaswami, 2020*; *McDiarmid et al., 2019*); nonetheless, a consensus model on the general principles underlying habituation is yet to emerge. Physiological and genetic work in *Aplysia* and *Caenorhabditis elegans* was consistent with a model in which homosynaptic depression of excitatory synapses drives habituation (*Bailey and Chen, 1983*; *Rose et al., 2003*; although see *Glanzman, 2009*). In contrast, work in the *Drosophila* olfactory and gustatory systems indicates that the potentiation of inhibitory neurons drives habituation rather than depression of excitatory connections (*Das et al., 2011*; *Paranjpe et al., 2012*; *Trisal et al., 2022*), and habituation to specific orientations of visual cues in mice is associated with the potentiation of neuronal activity and synapses in the visual cortex (*Cooke et al., 2015*), which requires GABAergic interneurons (*Kaplan et al., 2016*; *Hayden et al., 2021*). These studies are more consistent with a model in which the potentiation of inhibition, rather than depression of excitation, drives habituation (*Cooke and Ramaswami, 2020*).

Recently, we found that long-term habituation of the response of larval zebrafish to sudden pulses of whole-field darkness, or dark flashes (DFs), involves multiple molecularly independent plasticity processes that act to suppress different components of the behavioural response (*Randlett et al., 2019*). Similar behavioural, pharmacological, and genetic experiments have led to comparable conclusions in acoustic short-term habituation (*Nelson et al., 2023*) and habituation in *C. elegans* (*McDiarmid et al., 2019*; *McDiarmid et al., 2020*), indicating that habituation generally acts via multiple modular plasticity processes. These modules act to mute or shift behavioural responses to repeated stimuli. How and where these processes are implemented in the circuit, and how conserved or derived these processes are across species or paradigms remains to be determined. Here we have used a combination of high-throughput behavioural analyses, pharmacology, and $Ca^{2+}$ imaging to dissect DF habituation. Our results are consistent with a model in which habituation results from a multidimensional and distributed plasticity process, involving multiple independent molecular mechanisms. We propose that GABAergic inhibition is central to DF habituation, but how individual cell types and molecular events lead to behavioural adaptations during habituation will require targeted genetic and cellular approaches.

## Results

### Volumetric two-photon $Ca^{2+}$ imaging of habituation learning

When stimulated with a DF, larval zebrafish execute an O-bend response (*Figure 1A*). The O-bend is characterized by a strong body bend and a large turn that forms part of the phototactic strategy of larval zebrafish, helping them navigate towards lit environments (*Burgess and Granato, 2007*; *Chen and Engert, 2014*). When presented with repeated DFs, larvae habituate and reduce their responsiveness, remaining hypo-responsive for multiple hours (*Figure 1B*; *Randlett et al., 2019*).

To explore the circuit mechanisms leading to this form of habituation, we asked how individual neurons within the DF responsive circuit adapt to repeated DFs. We used a head-fixed paradigm to perform two-photon $Ca^{2+}$ imaging in larvae expressing nuclear-targeted GCaMP7f pan-neuronally. Imaging was performed with a resonant scanner and piezo objective, enabling us to cover a volume of $\approx 600 \times 300 \times 120$ μm (x,y,z) sampled at $0.6 \times 0.6 \times 10$ μm resolution, leading to the detection of $30890 \pm 3235$ Regions of Interest (ROIs) per larvae ($\pm$ SD, *Figure 1C–E*). ROIs were aligned to the Z-Brain atlas coordinates (*Randlett et al., 2015*), demonstrating that this volume spans the majority of the midbrain, hindbrain, pretectum, and thalamus (*Figure 1C–E*).

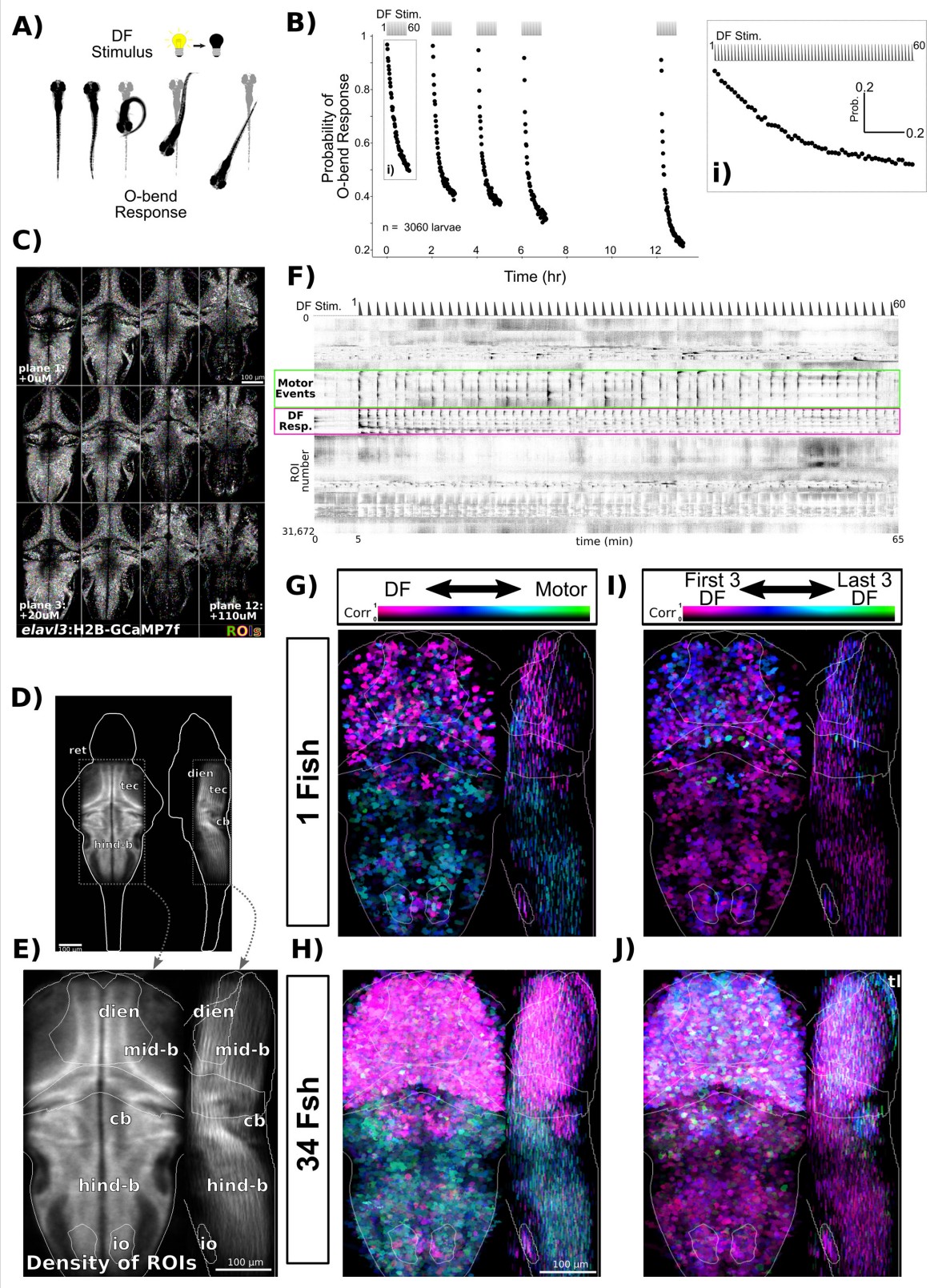

**Figure 1.** Volumetric two-photon $Ca^{2+}$ imaging of dark flash (DF) habituation. (**A**) In response to a DF, larval zebrafish execute a high-amplitude turn called an O-bend response. (**B**) Habituation results in a progressive decrease in response probability to DFs repeated at 1 min intervals, delivered in four blocks of 60 stimuli, separated by 1 hr of rest (from 0:00 to 7:00), and after a 5 hr retention period (12:00-). Inset (**i**) shows an expanded view of the first training block. (**C**) *Tg(elavl3:H2B-GCaMP7f)* larvae were imaged across 12 z-planes at 10 µm steps. Regions of Interest (ROIs) are overlaid in random

*Figure 1 continued on next page*

*Figure 1 continued*

colours. (**D**) Density of detected ROIs registered and plotted in the Z-Brain coordinate space. n = 1,050,273 ROIs across 34 larvae. (**E**) Cropped field of view used for plotting and analysing Ca$^{2+}$ imaging data and approximate anatomical localizations of major brain areas: dien, diencephalon; mid-b, midbrain; cb, cerebellum; hind-b, hindbrain; io, inferior olive; ret, retina; tec, tectum. (**F**) Functional responses of neurons to 60 DFs at 1 min intervals, plotted as a clustered heatmap ('rastermap'; *Pachitariu et al., 2017*, https://github.com/MouseLand/rastermap, copy archived at *MouseLand, 2023*) where rows represent individual neurons ordered by the similarities in their activity. Darker shades reflect increased activity. This clustering reveals neurons that are tuned to the DF stimuli (pink box) or motor events (green box). Dashed trace above the heatmap depicts the DF stimulus convolved with a kernel approximating H2B-GCaMP7f kinetics. (**G**) ROIs in an individual fish plotted based on their correlation and tuning to regressors defining either motor or DF stimulus events, highlighting the spatial distributions of these tunings across the imaged population. Plotted as a maximum intensity projection. (**H**) Same analysis as (**G**), but across the entire population of 34 larvae. (**I**) ROIs in an individual fish plotted based on their correlation and tuning to regressors defining either the first or last three DF stimuli. (**J**) Same analysis as (**I**), but across the entire population of 34 larvae. tl, torus longitudinalis.

The online version of this article includes the following figure supplement(s) for figure 1:

**Figure supplement 1.** Validation of motion analysis based on image artefacts during two-photon imaging.

We focused on a single training block of 60 DFs to identify neuronal adaptations that occur during the initial phase of learning (*Figure 1Bi*). This paradigm induced strong Ca$^{2+}$ activity in neurons (*Figure 1F*), some of which were clearly associated with the DF stimuli. Ca$^{2+}$ transients in response to DFs generally decreased across the 60 stimuli, though this pattern was not seen in all neurons, and substantial heterogeneity in their adaptations was observed. Strong correlated patterns were also seen in large groupings of neurons, predominantly in the hindbrain, which were associated with movement events through their correlation with motion artefacts in the imaging data (*Figure 1—figure supplement 1*).

To explore the spatial patterns in these data, we used a two-dimensional lookup table to visualize tuning with regressors representing either DF stimuli or movement (*Figure 1G and H*). This revealed segregated populations of neurons coding for the DFs (pink) and movement (green/teal). As expected, DF-tuned neurons were located predominantly in visual sensory areas of the midbrain (tectum) and the diencephalon (pretectum and thalamus). Motor-coding neurons dominated in the hindbrain, with the exception of the cerebellum and inferior olive, which was predominantly tuned to the sensory stimulus. Some neurons did show approximately equal correlation values to both stimuli, as evidenced by the bluish hues. Finally, some areas of the brain appeared to contain mixtures of neurons with different coding properties, including the ventral diencephalon and midbrain.

To determine if there was any spatial logic to how different neurons adapt their responsiveness to DFs during imaging, we plotted the ROIs using a lookup table highlighting the preference for either the first three DFs (pink, naive response) or last three DFs (green, trained response). Strong preferences for the naive stimuli reflect a depressing response profile (*Figure 1I and J*). While most neurons did show tuning consistent with strong depression, there were neurons that showed an equal preference for naive and trained stimuli, or even stronger preference for the latter, indicating stable or potentiating response profiles. These non-depressing neurons were mostly contained in the dorsal regions of the brain, including the torus longitudinalis, cerebellum, and dorsal hindbrain. These results demonstrate that while the majority of neurons across the brain depress their responsiveness during habituation, a smaller population of neurons exists that show the opposite pattern.

## Functional classification and anatomical localization of neuronal types observed during habituation learning

To explore the functional heterogeneity within the DF-tuned neurons, we used affinity propagation clustering. This method has the advantage that cluster numbers do not need to be defined beforehand and instead attempt to identify the most representative response profiles (*Förster et al., 2020*). This identified 12 clusters that differed both in their adaptation to repeated DFs, as well as the shape of their response to the DF (*Figure 2A and B*).

We therefore use these two aspects of the response to label the clusters:
Adaptation Profile.

No adaptation = $_{noA}$ : cluster 1, 9, and 10
Weak depression = $_{weakD}$ : clusters 5, 6, and 11
Medium depression = $_{medD}$ : clusters 2, 3, and 7

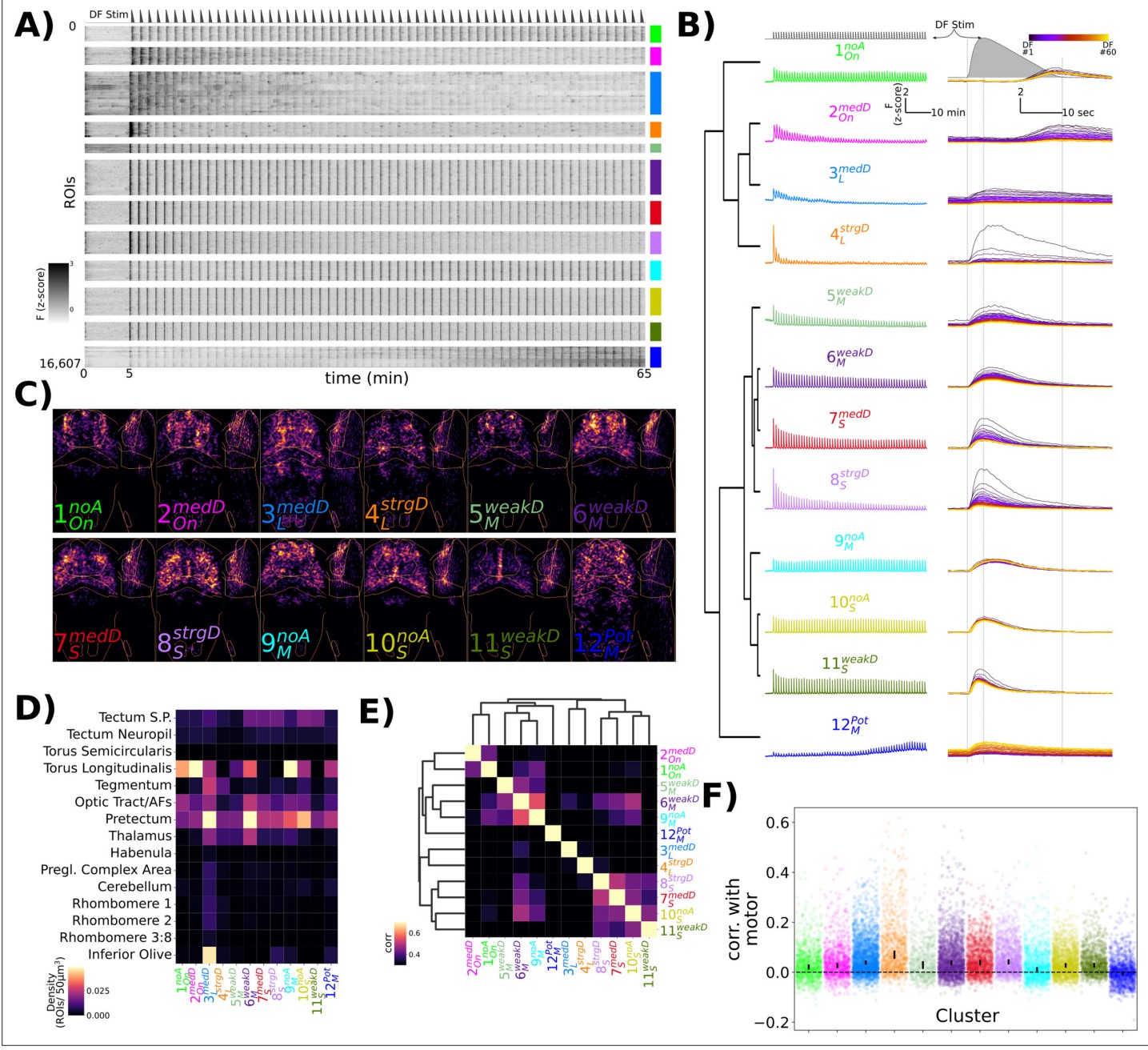

**Figure 2.** Characterization of functional response types during habituation learning. (**A**) Heatmap of the response profiles of ROIs categorized into 12 functional clusters. n = 16,607 ROIs from 34 larvae. (**B**) Average z-scored fluorescence of each functional cluster plotted for the whole experiment (left column) and centred on each dark flash (DF) stimulus (right column), demonstrating the differences in both *Adaptation Profiles* and *Response Shape* for each cluster. Clusters were identified using Affinity Propagation clustering (affinity = Pearson correlation, damping = 0.9, preference = -9), and organized using hierarchical clustering, distance = complete, correlation. Dashed lines in the top panels are the DF stimulus convolved with a kernel approximating H2B-GCaMP7f kinetics, used as the regressor in the analysis. (**C**) Summed intensity projection of the ROIs belonging to each functional cluster in Z-Brain coordinate space depicting their physical locations in the brain. Projection images are normalized to the maximum value. (**D**) Heatmap depicting the density of each cluster that is found within different Z-Brain regions. (**E**) Correlogram calculated from the Pearson correlation in downsampled volumes for the ROI centroid positions for each cluster (see 'Methods'). Hierarchical clustering, distance = complete, correlation. (**F**) Correlation between motor events and the Ca²⁺ traces for each ROI assigned to the functional clusters. dots = individual ROIs, bar height = 99.99999% confidence interval around the median value.

Strong depression = $strgD$ : clusters 4 and 8
Potentiation = $Pot$ : cluster 12

Response Shape.

On-response = $On$ : clusters 1 and 2
Long/sustained response = $L$ : clusters 3 and 4
Medium-length response = $M$ : clusters 5, 6, and 9
Short/transient response = $S$ : clusters 7, 8, 10, and 11

Yielding clusters: $1_{On}^{noA}$, $2_{On}^{medD}$, $3_{L}^{medD}$, $4_{L}^{strgD}$, $5_{M}^{weakD}$, $6_{M}^{weakD}$, $7_{S}^{medD}$, $8_{S}^{strgD}$, $9_{M}^{noA}$, $10_{S}^{noA}$, $11_{S}^{weakD}$, and $12_{M}^{Pot}$

While these results indicate the presence of a dozen functionally distinct neuron types, such clustering analyses will force categories upon the data irrespective if such categories actually exist. To determine if our cluster analyses identified genuine neuron types, we analysed their anatomical localization (*Figure 2C–E*). Since our clustering was based purely on functional responses, we reasoned that anatomical segregation of these clusters would be consistent with the presence of truly distinct types of neurons. Indeed, we observed considerable heterogeneity both within and across brain regions. For example, $11_{S}^{weakD}$ was mostly restricted to medial positions within the optic tectum; $3_{L}^{medD}$ and $4_{L}^{strgD}$ were more prevalent within motor-related regions of the brain including the tegmentum and hindbrain rhombomeres; $9_{M}^{noA}$ was the most prominent cluster in the torus longitudinalis, consistent with the presence of non-depressing signals in the area (*Figure 1I and J*).

We then quantified the similarity in the spatial relationships among the clusters by looking at the correlations in the positions of the ROIs in the Z-Brain (*Figure 2E*). This revealed similar hierarchical relationships to those identified functionally (*Figure 2B*), especially with respect to *Response Shape*, indicating that physical location is associated with functional response type.

Finally, since our functional analysis was performed purely based on correlations with the DF stimuli, we asked to what extent neurons belonging to each cluster were correlated with motor output (*Figure 2F*). This identified $4_{L}^{strgD}$ as the most strongly correlated to motor output, consistent with its strong habituation profile and its localization within motor regions of the hindbrain. This indicates that $4_{L}^{strgD}$ neurons likely occupy the most downstream positions within the sensorimotor network.

These results highlight a diversity of functional neuronal classes active during DF habituation. Whether there are indeed 12 classes of neurons or whether this is an over- or underestimate awaits a full molecular characterization. Independent of the precise number of neuronal classes, we proceed under the hypothesis that these clusters define neurons that play distinct roles in the DF response and/or its modulation during habituation learning.

## Pharmacological screening to identify habituation modulators

We next used a pharmacological screening approach to both identify molecular mechanisms of habituation and further probe the habituating circuit. For this, we screened 1953 small-molecule compounds with known targets (*Figure 3—source data 1*), in conjunction with the high-throughput assay we previously established, which has a maximum throughput of 600 larvae/day (*Figure 3A*; *Randlett et al., 2019*). As we aimed to identify modulators specific for habituation, we included additional behavioural assays as controls, including the response to acoustic stimuli, the optomotor response (OMR), and the spontaneous swimming behaviour of the fish in the absence of stimulation (*Figure 3B and C*). In each 300-well plate, 40 groups of six larvae were treated in individual wells and compared to 60 vehicle-treated controls (*Figure 3A*). We chose these numbers based on a sub-sampling analysis that determined these numbers were sufficient to identify the effect of a known modulator of habituation (haloperidol; *Randlett et al., 2019*) at a false-negative rate of less than 0.05 (not shown), while allowing us to screen 80 compounds per experiment across two plates.

We were able to collect the full behavioural record of 1761 compounds (*Figure 3D*, *Figure 3—source data 2*), indicating that the fish survived the treatment and maintained their ability to swim. Behavioural records for fish treated with each compound were compressed into a fingerprint (*Rihel et al., 2010*) – a vector representing the strictly standardized mean difference (SSMD) across 47 aspects of behaviour (see 'Methods'). For measurements related to DF habituation behaviour, responses were time-averaged across three epochs chosen to highlight the changes in habituation: the naive response (first five DFs), the response during the remaining training flashes, and the re-test block 5 hr after

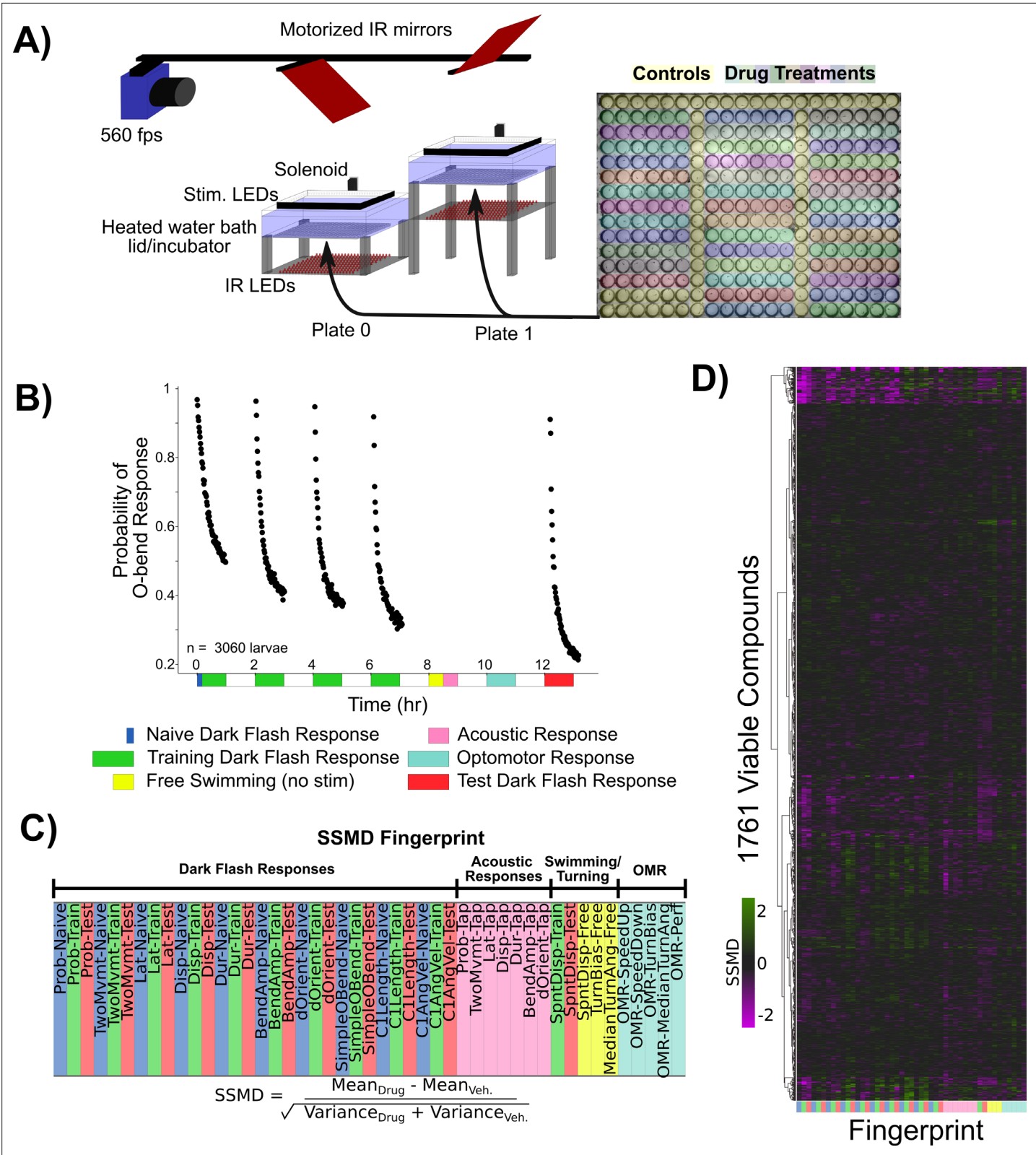

**Figure 3.** Pharmacological screening for dark flash habituation modulators. (**A**) Screening setup to record larval zebrafish behaviour in 300-well plates, which are placed below a 31°C water bath that acts as a heated lid for the behaviour plates. Two 300-well plates are imaged in alternation using mirrors mounted on stepper motors. Fish are illuminated with infrared LEDs and imaged with a high-speed camera recording at 560 frames per second (fps). Visual stimuli are delivered by a rectangular ring of RGB LEDs, and acoustic stimuli are delivered via a solenoid mounted on the back of the water

*Figure 3 continued on next page*

Figure 3 continued

tank. Colours overlaid on the 300-well plate indicate the arrangement of small-molecule treatments and controls (yellow). (**B**) Habituation results in a progressive decrease in responsiveness to dark flashes repeated at 1 mi intervals, delivered in four training blocks of 60 stimuli, separated by 1 hr of rest (from 0:00 to 7:00). This epoch is separated into periods reflective of the naive response (first five stimuli, blue), and the remaining 235 stimuli during training (green). From 8:00 to 8:30, no stimuli are delivered and fish are monitored for spontaneous behaviour (yellow). From 8:30 to 9:00, fish are given acoustic stimuli via the solenoid tapping on the water bath (pink). From 10:00 to 11:00, larvae are stimulated with alternating leftward and rightward motion using the RGB LEDs to induce the optomotor response and turning towards the direction of motion (light blue). Finally, at 12:00–13:00, larvae are given 60 additional dark flashes during the test period (red). Same data as *Figure 1B*. (**C**) The strictly standardized mean difference (SSMD) is calculated across these different time periods, behaviours, and the different components of O-Bend behavioural habituation (*Randlett et al., 2019*). All compounds were dosed at 10 µM in 0.1% DMSO (n = 6 larvae), relative to 0.1% DMSO vehicle controls (n = 60 larvae). (**D**) These vectors are assembled across all screened compounds that were viable and did not cause death or paralysis of the larvae. Displayed as a hierarchically clustered heatmap of behavioural fingerprints (vectors of SSMD values). Clustering distance = ward, standardized Euclidean.

The online version of this article includes the following source data for figure 3:

**Source data 1.** Small-molecule library, Selleckchem Bioactive: FDA-approved/FDA-like small molecules.

**Source data 2.** Behavioural fingerprint parameter descriptions.

**Source data 3.** Behavioural fingerprints for viable compounds.

---

training (*Figure 3B*). This was done across 10 different components of the DF response (probability of response, latency, displacement, etc.).

We found that 176 compounds significantly altered at least one aspect of measured behaviour, yielding a 9% hit rate (hit threshold of $|SSMD| \geq 2$). While the average effect was to suppress behavioural output ($\overline{SSMD} = -0.20$), which could reflect non-specific toxicity or a generalized inhibition of motor output, most small molecules induced both positive and negative changes in behavioural output, indicating that toxicity is not the primary phenotypic driver. While the false-negative rate is difficult to determine since so little is known about the pharmacology of the system, we note that of the three small molecules we previously established to alter DF habituation that were included in the screen – clozapine, haloperidol, and pimozide (*Randlett et al., 2019*) – the first two were identified among our hits while pimozide was lethal at the 10 µM screening concentration.

## Correlational structure in the pharmaco-behavioural space

To explore the pharmaco-behavioural space in our dataset, we clustered the hits based on their behavioural phenotypes (*Figure 4A*). This strategy can identify compounds that share common pharmacological targets or perhaps distinct pharmacological targets that result in convergent behavioural effects (*Bruni et al., 2016*; *Rihel et al., 2010*). Indeed, compounds known to target the same molecular pathways often showed similar behavioural fingerprints lying proximal on the linkage tree, indicating that our dataset contains sufficient signal-to-noise to recover consistent pharmaco-behaviour relationships.

Alternatively, compounds can be considered as tools to manipulate different aspects of brain function agnostic to their molecular mechanisms. Consequently, using similarities and differences among the induced alterations should uncover molecular and neural linkages among different behavioural outputs. Following this logic, the ability of a compound to co-modify different aspects of behaviour would reflect molecular and/or circuit-level dependencies. For example, visual behaviours that all depend upon photoreceptors should be similarly affected by any compounds that modulate phototransduction in these photoreceptors. We quantified these relationships by calculating the correlated effects on our different behavioural measurements across compounds (*Figure 4B*).

Consistent with our previous results highlighting uncorrelated learning across the behavioural components of the O-bend response during habituation (*Randlett et al., 2019*), we found that different aspects of the response were independently affected pharmacologically, resulting in distinctive correlated groupings within the correlogram. While we previously found that O-bend response probability and latency habituate independently in individual fish (*Randlett et al., 2019*), in our small-molecule screen data these appear to be tightly coupled (*Figure 4B*). The performance of the animals in the OMR assay under different treatments was also associated with O-bend probability and latency, suggesting that pharmacological modulation of vision or arousal could drive these correlations within the small-molecule screen dataset.

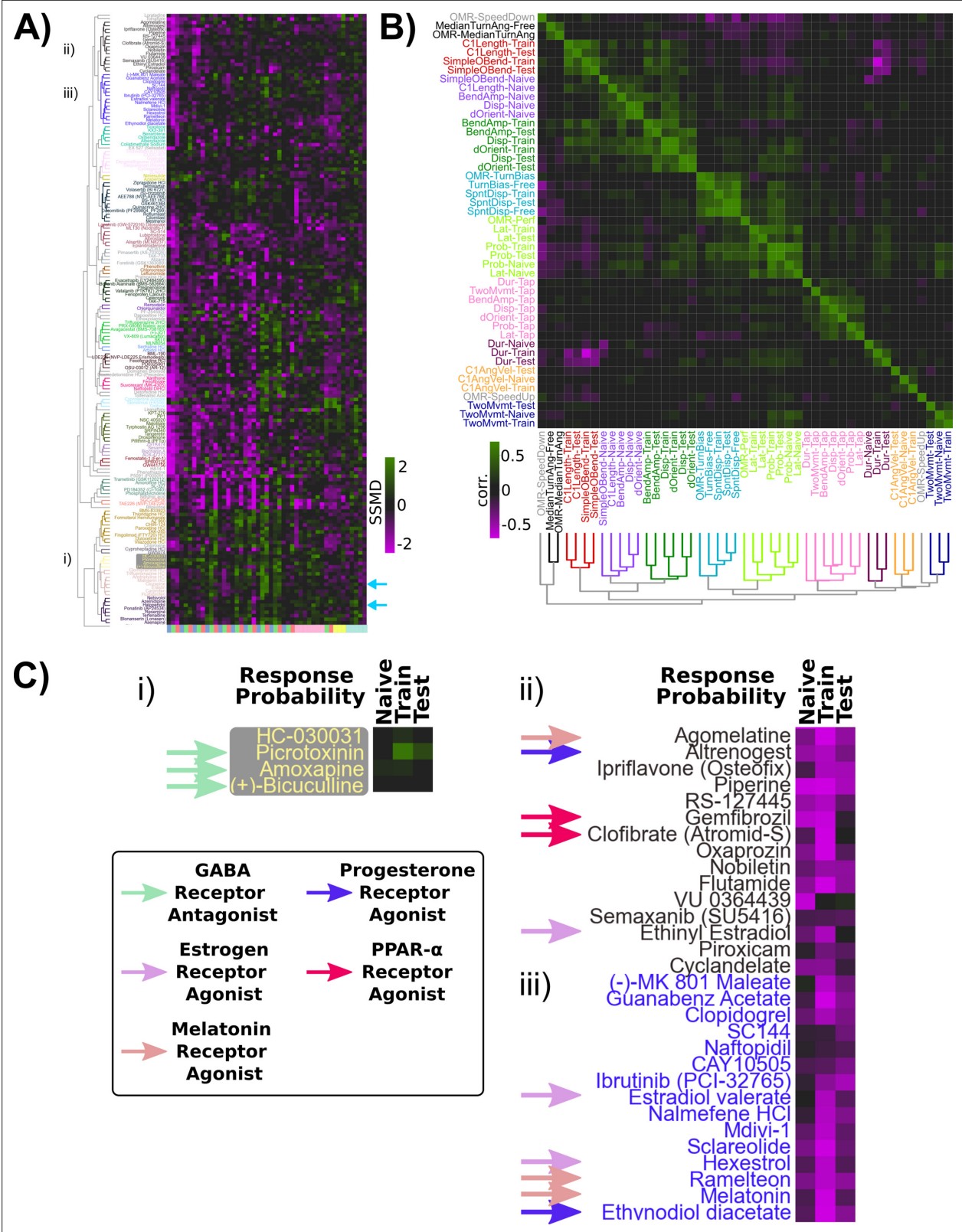

**Figure 4.** Pharmaco-behavioural analyses of behaviour-modifying compounds. (**A**) Clustered heatmap of the behavioural fingerprints for the 176 hits of the screen, showing at least one behaviour measure with $|SSMD| \geq 2$. Clustering distance = ward, standardized Euclidean, colour/cluster threshold = 9.5. This led to the re-identification of haloperidol and clozapine as habituation modifiers (light blue arrows). (**B**) Clustered correlogram of the Pearson correlation coefficients for the different measured components of behaviour across hits (same data as **A**) revealing the independence

*Figure 4 continued on next page*

*Figure 4 continued*
or co-modulation of behaviours. Clustering distance = average, correlation, colour/cluster threshold = 1.5. (**C**) Subsets of clustered heatmap from (**A**), highlighting the similar phenotypes exhibited by (**i**) GABA receptor antagonists and (**ii**, **iii**) melatonin receptor agonists, oestrogen receptor agonists, progesterone receptor agonists, and peroxisome proliferator-activated receptor alpha (PPARα) agonists. Heatmap is cropped to the first three columns of (**A**), depicting the strictly standardized mean difference (SSMD) of response probability relative to vehicle controls.

These analyses confirm habituation behaviour manifests from multiple distinct molecular mechanisms that independently modulate different behavioural outputs.

## Modulation of habituation by GABA, melatonin, and oestrogen signalling

For the remainder of the analyses, we decided to focus on the mechanisms leading to the habituation of response probability as this is the criterion for which it is easiest to identify the link between neural activity and behaviour, providing the best entry point for studying the circuit mechanisms of long-term habituation. To identify the most promising hits, we sought to identify compounds that:

1. Have minimal effects on the naive response to DFs, but strong effects during the training and/or memoryretention periods. This would prioritize pathways that affect habituation, rather than simply DF responsiveness.
2. Have minimal effects on other aspects of behaviour, in order to exclude compounds that would alter generalized arousal, movement ability/paralysis, or visual impairment. Such compounds would strongly influence DF responsiveness, but likely independently of pathways related to habituation.
3. Show similar behavioural effects to other compounds tested that target the same molecular pathway. Such relationships can be used to cross validate, yet we note that our library choice was very broad, and target coverage is non-uniform. Therefore a lack of multiple hits targeting the same pathway should not be taken as strong evidence of a false positive.

This manual prioritization led to the identification of the GABA$_{A/C}$ receptor antagonists bicuculline, amoxapine, and picrotoxinin (PTX). PTX treatment had the strongest effects, with increased responsiveness to DFs during the training and test periods, indicative of defects in habituation (*Figure 4Ci*). Dose–response experiments confirmed a strong effect of PTX on inhibiting the progressive decrease in responsiveness during habituation learning at 1–10 µM doses (*Figure 5A*). Importantly, like the naive DF response, the probability of responding to an acoustic stimulus and the optomotor response (OMR) was not inhibited (*Figure 5—figure supplement 1A*). While strong GABA$_{A/C}$R inhibition results in epileptic activity in larval zebrafish, we did not observe evidence of seizure-like behaviour at these doses, consistent with a partial GABA$_{A/C}$R in our experiments and previous results (*Bandara et al., 2020*). Therefore, we conclude that partial antagonism of GABA$_A$R and/or GABA$_C$R is sufficient to strongly suppress habituation but not generalized behavioural excitability, indicating that GABA plays a very prominent role in habituation. This is consistent with a model in which the potentiation of inhibition actively silences sensory-induced activity during habituation to suppress motor output (*Cooke and Ramaswami, 2020*; *Ramaswami, 2014*).

We next turned our attention to the upper portion of the clustered behavioural fingerprint graph (*Figure 4A*), where strong and relatively specific inhibition of responsiveness during training and testing were observed, indicative of enhanced habituation (*Figure 4Cii, iii*). Among the hits observed here were multiple agonists of both melatonin and oestrogen receptors, indicating that hormonal signalling may play a prominent role in habituation. Dose–response studies with melatonin confirmed strong potentiation of habituation (*Figure 5B*). Melatonin did cause a decrease in spontaneous movement behaviour, consistent with its role in arousal/sleep regulation in zebrafish and other vertebrates (*Gandhi et al., 2015*; *Zhdanova et al., 2001*), yet melatonin did not inhibit the naive response to DFs, the responsiveness to acoustic stimuli or OMR performance (*Figure 5B*, *Figure 5—figure supplement 1B*). Melatonin's effect on habituation was also most prominent for the probability of response and did not strongly alter habituation for displacement (*Figure 5—figure supplement 1F*), indicating that it does not cause generalized sedation but modulates specific aspects of behaviour at these doses, including increasing habituation of the probability of response.

We similarly validated that the oestrogen receptor agonists ethinyl estradiol and hexestrol potentiated habituation at 5–100 µM and 1–10 µM doses, respectively (*Figure 5C and D*). Ethinyl estradiol

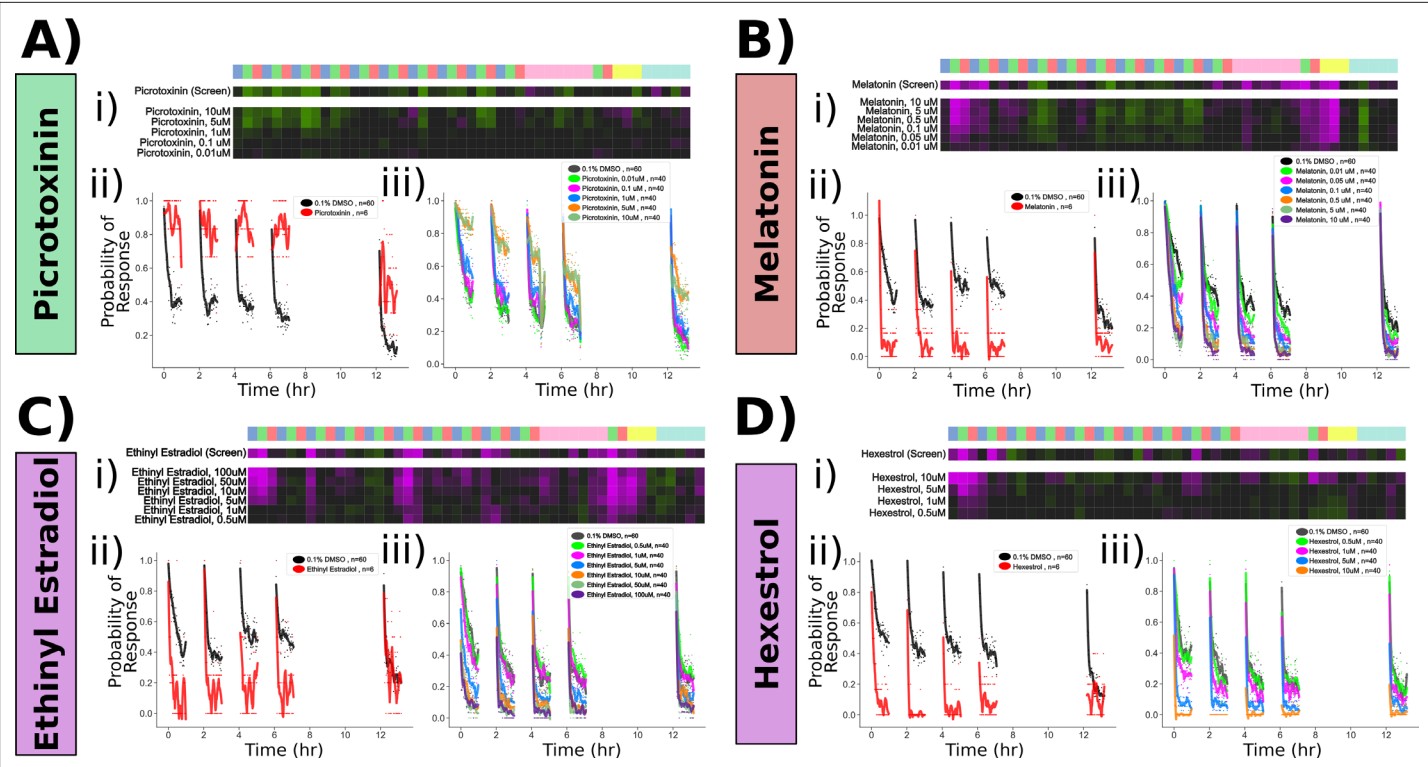

**Figure 5.** Confirmed pharmacological modulators of habituation. Dose–response studies for (**A**) picrotoxinin, (**B**) melatonin, (**C**) ethinyl estradiol, and (**D**) hexestrol. Displayed for each treatment are (**i**) behavioural fingerprint for the original screen data (10 uM) and the dose–response data. (**ii**) Original screen data for the probability of response to dark flash (DF) stimuli. Each dot is the probability of response to one flash. Lines are smoothed in time with a Savitzky–Golay filter (window = 15 stimuli, order = 2). (**iii**) Dose–response data for the probability of response, plotted as in (**ii**).

The online version of this article includes the following figure supplement(s) for figure 5:

**Figure supplement 1.** Pharmacological manipulation of control behaviours and response displacement during habituation.

strongly suppressed movement rates at these doses, and both treatments suppressed acoustic responsiveness and OMR performance at doses ≥ 10 µM (*Figure 5—figure supplement 1C and D*). Thus, it is less clear how specific or generalized oestrogen receptor agonism is on behaviour, although the effective doses of hexestrol for influencing habituation (1–5 uM) were lower than those that significantly affected other behaviours (10 uM). Nevertheless, we decided to focus on PTX and melatonin for the remaining experiments.

Our screening approach identified both expected (GABA) and unexpected (melatonin, oestrogen) pathways that strongly modulate habituation of responsiveness. We also implicated other pathways in habituation, including progesterone and PPARα (*Figure 4C*), and identified compounds that strongly modify other aspects of behaviour (OMR, acoustic and spontaneous behaviour). These hits can be mined for future projects investigating the molecular basis of behaviour.

## Pharmacological manipulations of functional circuit properties during habituation

Our Ca$^{2+}$ imaging experiments identified 12 distinct functional classes of neurons during habituation learning, but it is unclear how these might be organized in a circuit. Based on the diversity of functional response profiles identified, it is clear that solving this circuit will take considerable further work. As a starting point in this long-term effort, we used the pharmacological manipulations as these treatments provide us with tools to ask how treatments that potently alter habituation behaviour also alter the functional properties of neurons. We compared the Ca$^{2+}$ activity patterns after treatment with vehicle (0.1% DMSO), PTX, or melatonin (*Figure 6*). At the behavioural level, we found a trend indicating that we were able to manipulate habituation pharmacologically in our tethered imaging assay, though this was very subtle (*Figure 6A*). This discrepancy relative to the very strong behavioural

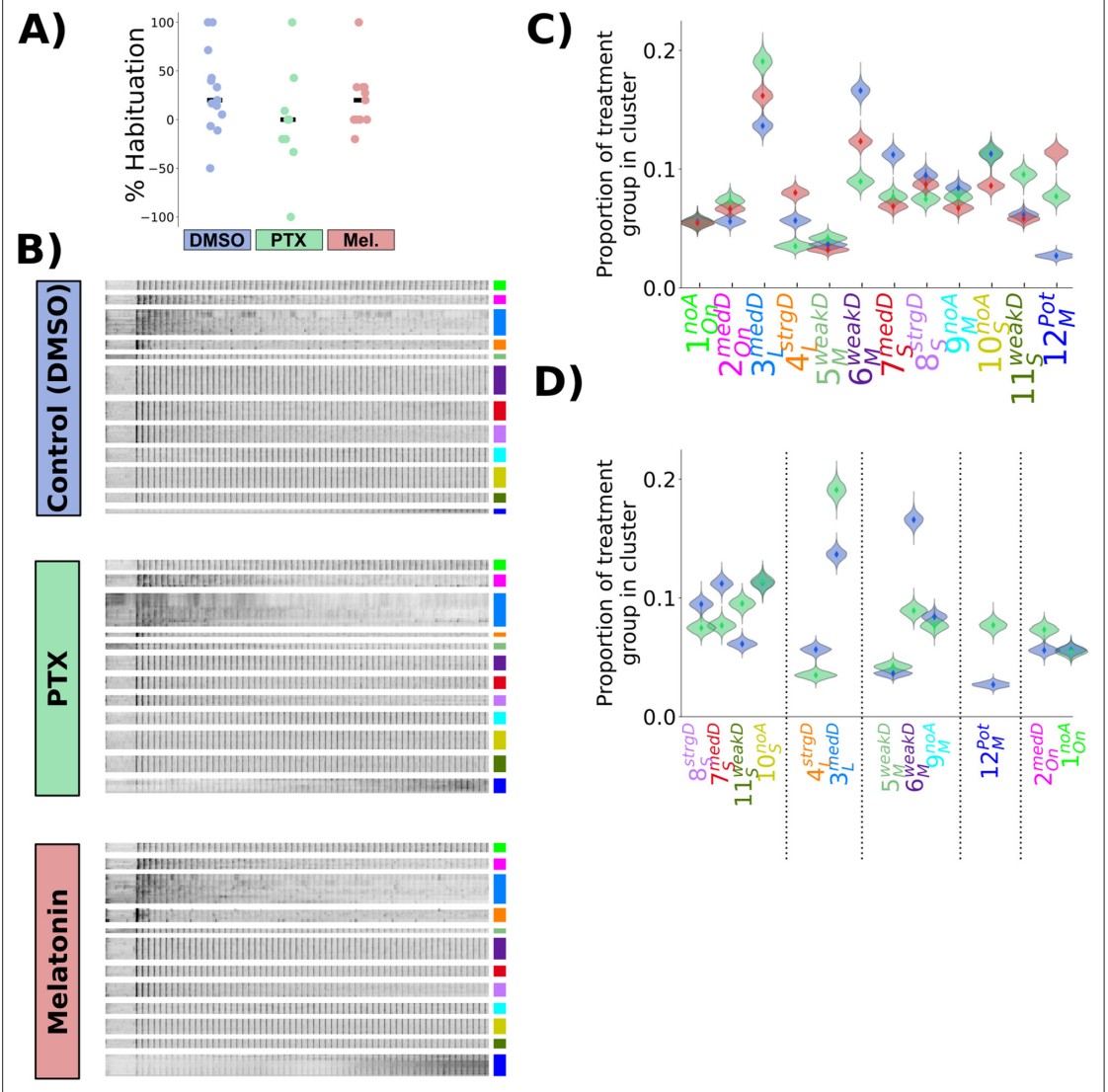

**Figure 6.** Picrotoxinin and melatonin alter the proportions of functionally identified neurons. (**A**) Percent habituation for larvae during Ca²⁺ imaging, calculated as $\%Habituation = 100 \times (1 - \frac{P(Resp_{31\to60})}{0.5 \times (P(Resp_{1\to30}) + P(Resp_{31\to60}))})$ (**B**) Heatmap of response profiles of ROIs categorized into the 12 functional clusters from larvae treated with DMSO (vehicle control, n = 428,720 total ROIs in 14 larvae), Picrotoxinin (PTX, 10 uM, n = 271,037 total ROIs in 9 larvae), or melatonin (1 uM, n = 350,516 total ROIs in 11 larvae). (**C**) Proportion of neurons belonging to each functional cluster across treatment groups. Distributions for violin plots are bootstrapped from 5000 replicates. (**D**) Same data as (**C**), only showing the data for PTX vs DMSO vehicle control, reordered to reflect the cluster Adaptation Profiles grouped by cluster Response Shape.

The online version of this article includes the following figure supplement(s) for figure 6:

**Figure supplement 1.** Mean response of functionally identified clusters after different pharmacological treatments.

effects in freely swimming animals (*Figure 5*) likely results from the head-restrained protocol, which itself strongly inhibits behavioural output. Yet, since we did observe a trend in behavioural data, we proceeded under the assumption that the compounds were having the desired effects.

As PTX and melatonin have opposing effects on habituation behaviour, we reasoned that these two treatments should have opposite effects in the circuit, with PTX inhibiting depression and melatonin promoting depression. Indeed, melatonin has been found to increase the effects of GABA, and so such a relationship could be direct (*Cheng et al., 2012*; *Niles et al., 1987*). In contrast to this straightforward hypothesis, what we observed was considerably more complex. We did not observe alterations of the average response profiles of individual neuronal classes, which remained indistinguishable after the treatments (*Figure 6—figure supplement 1C-D*). Instead, the

proportion of neurons that belonged to the different classes was altered (**Figure 6B–D**). Therefore, the pharmacological manipulations did not alter the activity of neurons in such a way as to alter the average activity states of populations, but instead the proportion of neurons belonging to different populations changed. This may point to fixed and relatively inflexible processing strategies that the brain is using in the context of DF habituation which constrain the possible functional response types.

The effect of PTX on cluster reassignment generally tended towards weaker depression, increasing the proportion of cells falling into the weaker depressing classes at the expense of strongly depressing classes for a given response shape (**Figure 6D**). This pattern was most clear in the classes with 'short' and 'long' *Response Shapes*, which are those that included the most strongly depressed classes of neurons.

Based on the hypothesis that melatonin and GABA cooperate during habituation, we expected PTX and melatonin to have opposite effects. This clearly does not fit with our observations: for example, the size of the $12_M^{Pot}$ neuron population was increased by both PTX and melatonin (**Figure 5C**). While habituation of the probability of response is oppositely modulated by PTX and melatonin, this is not true of behaviour globally – the behavioural fingerprints of melatonin and GABA are not opposites (**Figure 5A and B**) and opposing effects are not seen for the habituation of displacement (**Figure 5— figure supplement 1E and F**). Therefore, a lack of coherent shifts across the entire neural population when applying these treatments is expected. However, opposite effects of PTX and melatonin were observed for $4_L^{strgD}$ neurons (**Figure 6C**), which we found to be most strongly correlated with motor output (**Figure 2F**). Therefore, this class might be most critical for habituation of response probability.

Combined, these experiments reveal that pharmacological manipulations that affect habituation behaviour manifest in complex functional alterations in the circuit. These effects cannot be captured by a simple model, and considerable additional knowledge of the circuit, including the connectivity and signalling capacity of different neurons, will be necessary to understand these dynamics.

## Identification of GABAergic neuron classes in the habituating circuit

Since our pharmacological experiments point to the importance of GABAergic inhibition in habituation, we asked which functional classes of neurons are GABAergic? An obvious model for habituation would assign a GABAergic identity to the $12_M^{Pot}$ neurons that potentiate their responses, and thus could act to progressively depress the responses of other neuronal classes. We began with virtual co-localization analyses with 3D atlases to identify candidate molecular markers for functionally identified neurons. Such a strategy can be powerful to generate hypotheses from brain-wide imaging data, provided sufficient stereotypy exists in the positioning of neurons, and the relevant marker exists in the atlas (**Dunn et al., 2016**; **Randlett et al., 2015**). Therefore, we analysed the spatial correlations for markers contained in the Z-Brain (**Randlett et al., 2015**), Zebrafish Brain Browser (**Gupta et al., 2018**; **Marquart et al., 2017**; **Tabor et al., 2018**), and mapZebrain atlases (**Kunst et al., 2018**; **Shainer et al., 2022**). We identified markers showing the highest spatial correlations with any of our functional clusters (corr. > 0.15, n = 89 of 752 markers) and organized these hierarchically (**Figure 7A**). GABAergic reporter lines based on the *gad1b* promoter were located in a region of the hierarchy showing greatest spatial similarity with $10_S^{noA}$ and $11_S^{weakD}$(**Figure 7B–E**). An enrichment along the medial tectum is common to markers in this region of the hierarchy, where the highest density of GABAergic neurons within the tectum resides.

To confirm that $10_S^{noA}$ and $11_S^{weakD}$ classes are GABAergic, we imaged the response of neurons in *Tg(Gad1b:DsRed); Tg(elavl3:H2B-GCaMP6s)* double transgenic larvae and classified neurons as *gad1b*-positive or -negative based on DsRed/GCaMP levels (**Figure 7F and G**). Indeed, we saw a heterogeneous distribution of *gad1b*-positive neurons across functional clusters, including a significant enrichment in not only $10_S^{noA}$ and $11_S^{weakD}$, but also the other two clusters with the 'short' *Response Shape* ($7_S^{medD}$ and $8_S^{strgD}$). The remaining clusters either showed no significant bias, indicating that they contain mixed populations, or a significant depletion of *gad1b*-positive cells, suggesting that they comprise mostly of excitatory or neuromodulatory neurons ($3_L^{medD}$ and $12_M^{Pot}$).

These experiments indicate that GABAergic neurons in the habituating circuit are not characterized by their *Adaptation Profile* (other than non-potentiating), and instead have a characteristic 'short' *Response Shape*, perhaps reflecting a transient bursting style of activity relative to other neuronal types that exhibit more sustained firing patterns. This lack of coherence in adaptation profile may

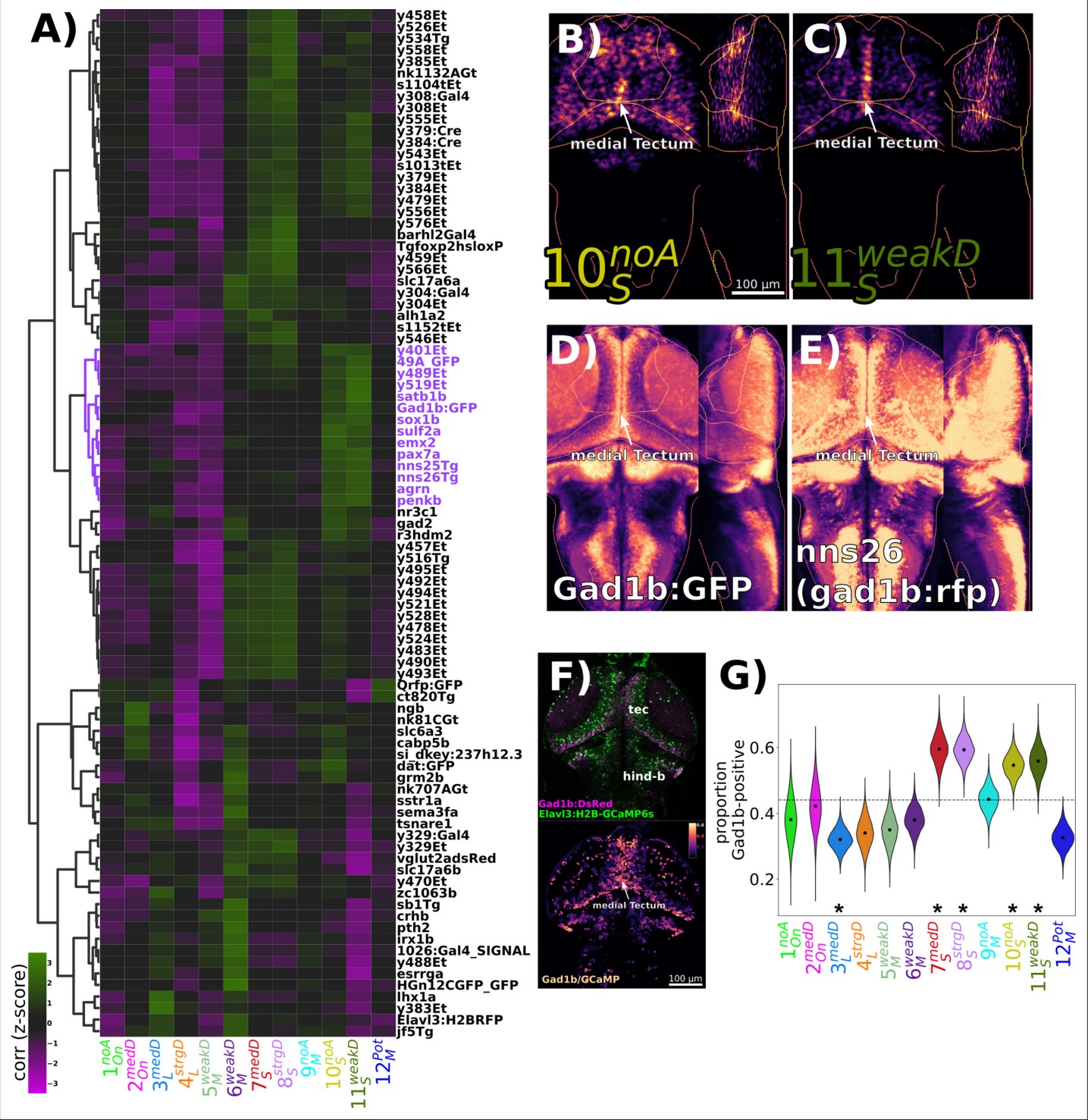

**Figure 7.** Identification of GABAergic neuronal classes. (**A**) Hierarchically clustered heatmap depicting the correlation of markers aligned to the Z-Brain atlas with the spatial arrangement of the 12 functional clusters (distance = complete, correlation). Correlation values are z-scored by rows to highlight the cluster(s) most strongly correlated or anti-correlated with a given marker. The subset of the hierarchy containing the *gad1b*-reporters is coloured in purple. (**B–D**) Normalized summed intensity projections of (**B**) $10_S^{noA}$, (**C**) $11_S^{weakD}$, (**D**) *TgBAC(gad1b:GFP)* (**Satou et al., 2013**), Z-Brain Atlas, and (**E**) *nns26, aka TgBAC(gad1b:LOXP-RFP-LOXP-GFP)* (**Satou et al., 2013**), mapZebrain Atlas. (**F**) Two-photon imaging of *Tg(Gad1b:DsRed);Tg(elavl3:H2B-GCaMP6s)* larvae depicting the raw data for each channel (top), and the ratio of Gad1b/GCaMP6s fluorescence in each ROI functionally identified using suite2p. (**G**) ROIs imaged in double transgenic larvae are assigned a cluster identity based on their correlation to the cluster mean trace and classified as Gad1b-positive based on a DsRed/GCaMP6s ratio of greater than 0.25. Dotted line = expected proportion based on total number of cells classified

*Figure 7 continued on next page*

*Figure 7 continued*

as Gad1b-positive. *p<0.05, Chi-square test with Bonferroni correction. Distributions for violin plots calculated by bootstrapping 5000 replicates. n = 1835 ROIs in six larvae.

explain why global manipulations of GABAergic signalling through PTX have complex manifestations in the functional properties of neurons (*Figure 6D*).

## Discussion
### Molecular mechanisms of DF habituation

To explore the molecular mechanisms of habituation, we performed a small-molecule screen testing for effects on DF habituation behaviour. Analyses of the correlated effects of drugs across different aspects of behaviour (*Figure 4*) are consistent with our previous results, indicating that habituation results from multiple molecularly independent plasticity processes which act to adapt different aspects of the DF response during habituation (*Randlett et al., 2019*). Here we focused our analysis on those pharmacological agents and pathways that strongly and relatively specifically modulated habituation when measuring response probability. We found that inhibition of GABA$_{A/C}$ receptors using PTX reduced habituation learning. GABA is the main inhibitory neurotransmitter in the zebrafish brain, and deficits in GABA signalling lead to epileptic phenotypes (*Baraban et al., 2005*). We were fortunate that our screening concentration (10 μM) did not cause seizures, but was still sufficient to inhibit habituation. This implies that the habituation circuit is exquisitely sensitive to changes in GABA signalling at levels well below the threshold required to drastically change excitatory-inhibitory balances. We cannot rule out the possibility that off-targets of PTX or subtle non-specific changes in excitatory/inhibitory balance alter habituation behaviour. However, the lack of strong modulation of other behaviours, including the response to acoustic stimuli or the optomotor response (*Figure 5—figure supplement 1A*), suggests that GABAergic inhibition plays a crucial role in the process of DF habituation.

A critical role for GABA in habituation is also consistent with data from *Drosophila*, where both olfactory and gustatory habituation have been linked to GABAergic interneurons (*Das et al., 2011*; *Paranjpe et al., 2012*; *Trisal et al., 2022*). Therefore, this circuit motif of increasing inhibition to drive habituation may be a conserved feature of habituation, which would allow for a straightforward mechanism for habituation override during dishabituation via dis-inhibition (*Cooke and Ramaswami, 2020*; *Trisal et al., 2022*).

Our screen also identified that neurohormonal signalling is critical for habituation, where melatonin and oestrogen receptor agonists potently increase habituation learning rate. The role of oestrogens in learning and memory is well established (*Luine et al., 1998*; *Nilsson and Gustafsson, 2002*). Though its role in habituation is less well explored, it has previously been shown to increase memory retention for olfactory habituation in mice (*Dillon et al., 2013*). To our knowledge, melatonin has not previously been implicated in habituation, though it has been implicated in other learning paradigms (*El-Sherif et al., 2003*; *Jilg et al., 2019*). Notably, melatonin was shown to block operant learning at night in adult zebrafish (*Rawashdeh et al., 2007*), and therefore melatonin appears to be able to both promote or inhibit plasticity in zebrafish, depending on the paradigm.

While melatonin and oestrogen were not strong candidates for involvement in DF habituation plasticity before our screen, their previous associations with learning and memory reinforce the idea that these molecules play critical roles in plasticity processes. In support of this idea, we have previously shown that habituation is regulated in a circadian-dependent manner (*Randlett et al., 2019*), and both melatonin and oestrogen levels fluctuate across the circadian cycle (*Alvord et al., 2022*; *Gandhi et al., 2015*; *Zhdanova et al., 2001*), suggesting that either or both of these pathways may act to couple the circadian rhythm with learning performance.

Finally, approximately 2% of the US population use melatonin as a sleep aid (*Li et al., 2022*), and a substantial proportion of US women take oestrogen as part of either oral contraceptives or hormone replacement therapy. Therefore, understanding the roles these molecules play in neuroplasticity is a clear public health concern.

## Circuit mechanisms of DF habituation

Based on behavioural experiments, we previously postulated that multiple plasticity loci cooperate in the habituating DF circuit, arranged in both parallel and series within the circuit (*Randlett et al., 2019*). Here, our Ca²⁺ imaging experiments identified a diverse range of neuronal *Adaptation Profiles*, including non-adapting and potentiating neurons spread throughout sensory- and motor-related areas of the brain. Thus, non-habituated signals are transmitted throughout the brain, consistent with a distributed learning process. Such a model is further supported with brain-wide imaging data for short-term habituation to looming stimuli, where distributed neurons were identified that showed differential rates of habituation (*Marquez-Legorreta et al., 2022*). It is important to point out that Marquez-Legorreta et al. did not observe non-adapting or potentiating neurons in their experiments. This may be due to the differences in analysis methods or could highlight a difference between short- and long-term habituation circuit mechanisms, the latter of which may rely on more complex circuit mechanisms involving both potentiation and suppression of neuronal responses.

We also observed classes exhibiting an On-response profile ($1_{On}^{noA}$ and $2_{On}^{medD}$). These neurons fire at the ramping increase in luminance after the DF, making it unlikely that they play a role in aspects of acute DF behaviour we measured here. These neurons exist in both non-adapting and depressing forms, suggesting a yet unidentified role in behavioural adaptation to repeated DFs.

While we have insufficient anatomical data to constrain circuit connectivity models that drive DF habituation, here we demonstrate the use of pharmacology, functional imaging, and neurotransmitter classifications to constrain our models. Specifically, pharmacology indicated a role for GABA and melatonin in habituation, and our functional imaging identified distinct classes of neuronal types in the DF circuit, including potentiating neurons ($12_{M}^{Pot}$). These results point to a model where $12_{M}^{Pot}$ neurons are GABAergic and thus progressively inhibit the other neuronal classes, and that perhaps this effect is bolstered by melatonin. However, in silico co-localization analyses and double transgenic Ca²⁺ imaging identified $12_{M}^{Pot}$ neurons as predominantly non-GABAergic, inconsistent with this simple model. Instead, we found that the GABAergic neurons in the circuit are characterized by their short burst of activity to the stimulus onset. If the GABAergic neurons are not increasing in their firing rates but do drive habituation, then perhaps it is the potentiation of GABAergic synapses that drives habituation. This is a somewhat unexpected model as studies of long-term synaptic plasticity (e.g. LTP and LTD) have overwhelmingly focused on plasticity at excitatory synapses. Although a functional link to behaviour is less well established, long-term inhibitory synaptic plasticity has been well documented, including inhibitory (i)-LTP and i-LTD (*Castillo et al., 2011*). Alternatively, there may be a key minority subset of $12_{M}^{Pot}$ neurons that are GABAergic and exert a strong influence over the rest of the circuit driving depression and habituation.

We also found that the same pharmacological treatments that result in strong alterations to habituation behaviour in freely swimming larvae (*Figure 5*) resulted in relatively subtle and complex functional alterations in the circuit (*Figure 6*). Making direct comparisons between freely swimming behaviour and head-fixed Ca²⁺ imaging is always challenging due to the differences in behaviour observed in the two contexts, and therefore our failure to identify a clear logic in these experiments may have technical explanations that will require approaches to measure neural activity from unrestrained and freely behaving animals to resolve (*Kim et al., 2017*). Alternatively, these results are again consistent with the idea that habituation is a multidimensional and perhaps highly non-linear phenomenon in the circuit, which cannot be captured by a simple model.

## Circuit loci of DF habituation

Where in the brain does habituation take place? As discussed above and previously, our data is inconsistent with a single locus of plasticity (*Randlett et al., 2019*). Instead, we propose that plasticity is distributed throughout the circuit. Since PTX inhibits most aspects of habituation learning (*Figure 5Ai*), these all may involve GABAergic motifs. Moreover, the different functional classes of neurons are distributed through sensory- and motor-related areas of the brain, consistent with the notion that habituation plasticity occurs in a very distributed manner. While distributed, there are clear associations between anatomical location and functional neuron type (*Figure 2A–E*), indicating that there is some degree of regional logic to the localization of *Adaptation Profiles*. For example, $5_{M}^{weakD}$ and $6_{M}^{weakD}$ are the most prevalent in the pretectum and mostly absent from the tegmentum and

posterior hindbrain, whereas $3_L^{medD}$ and $4_L^{strgD}$ are numerous in tegmentum and posterior hindbrain, and thus likely occupy more downstream positions in the sensorimotor circuit.

The tectum is one of the largest brain areas in larval zebrafish and is directly innervated by nearly all retinal ganglion cells (*Robles et al., 2014*). Therefore, the tectum is a prime candidate for implementing DF habituation for anatomical reasons. In further support of this notion, the neurons we have identified as GABAergic and propose to be driving habituation ($7_S^{medD}$, $8_S^{strgD}$, $10_S^{noA}$, and $11_S^{weakD}$) are concentrated in the tectum (*Figure 2C and D*). The tectum contains multiple anatomically distinct types of GABAergic neurons, most of which are locally projecting interneurons (SINs - superficial interneurons, ITNs - intertectal commisural neurons, and PVINs - periventricular interneurons), although GABAergic projection neurons have been observed with axons projecting to the anterior hindbrain (*Gebhardt et al., 2019*; *Martin et al., 2022*; *Nevin et al., 2010*; *Robles et al., 2011*). Therefore, we expect that our GABAergic classes correspond to subsets of these GABAergic tectal neurons, which is testable using genetic approaches based on marker co-expression and/or single-cell morphometric and transcriptomic analyses.

Beyond the tectum, conspicuous neuronal clustering was observed in the inferior olive and cerebellum, which have been implicated in motor-related learning behaviours in larval zebrafish (*Ahrens et al., 2012*; *Lin et al., 2020*; *Markov et al., 2021*). Both structures contained many stimulus-tuned neurons (*Figure 1I*), and non-adapting ($1_{On}^{noA}$, $9_M^{noA}$ and $10_S^{noA}$) and potentiating ($12_M^{Pot}$) neurons were among the most concentrated in the cerebellum (*Figure 2C and D*). Non-adapting $9_M^{noA}$ neurons were also prominent in the torus longitudinalis. The torus longitudinalis has recently been implicated in the binocular integration of luminance cues (*Tesmer et al., 2022*), and therefore is ideally placed to influence habituation to whole-field stimuli like DFs.

Collectively, our brain-wide imaging data indicate that the adaptations underlying habituation span many regions of the brain, and therefore a comprehensive model will need to span many regions of the brain in order to explain the neural and behavioural dynamics underlying habituation learning.

## Conclusion

Habituation is the simplest form of learning, yet despite its presumed simplicity a model of how this process is regulated in the vertebrate brain is still emerging. Here we have combined two methods offered by the larval zebrafish model: whole-brain functional imaging and high-throughput behavioural screening. By applying these methods to long-term habituation, we identified and validated pharmacological agents that strongly modulate habituation learning, and distinct classes of neurons that are activated by DFs and adapt their activity during learning. The systematic datasets we generated contain large amounts of additional information that await future validation and integration into our understanding of DF habituation. Nonetheless, the diversity of molecular pathways and functional neuronal types we have identified here indicates that considerable biological complexity exists that awaits discovery within the 'simplest' form of learning.

## Methods

### Animals

All experiments were performed on larval zebrafish at 5 days post fertilization (dpf), raised at a density of ≈1 larvae/mL of E3 media in a 14:10 hr light/dark cycle at 28–29°C. Wild-type zebrafish were of the TLF strain (ZDB-GENO-990623-2). Transgenic larvae used were of the following genotypes: *Tg(elavl3:H2B-GCaMP7f)^jf90* (*Yang et al., 2021*), *Tg(elavl3:H2B-GCaMP6s)^jf5* (*Freeman et al., 2014*), and *Tg(gad1b:DsRed)^nns26* (*Satou et al., 2013*). Zebrafish were housed, cared for, and bred at the Harvard MCB, UPenn CDB, and Lyon PRECI zebrafish facilities. All experiments were done in accordance with relevant approval from local ethical committees at Harvard University, the University of Pennsylvania, and the University of Lyon.

### High-throughput screening setup and protocol

Larvae were assayed for behaviour in 300-well plates using the apparatus described previously (*Randlett et al., 2019*). Briefly, each well is 8 mm in diameter and 6 mm deep, yielding a water volume of ≈300 uL. Behaviour plates are suspended below a water bath kept at 31°C, which acts as a heated lid to prevent condensation and maintains the water temperature in the well at 29°C.

Behaviour was tracked using a Mikrotron CXP-4 camera, Bitflow CTN-CX4 frame grabber, illuminated with IR LEDs (TSHF5410, digikey.com). Visual stimuli were delivered via a ring of 155 WS2812B RGB LEDs (144LED/M, aliexpress.com). For a DF stimulus, the LEDs were turned off for 1 s, and then the light intensity was increased linearly to the original brightness over 20 s. The optomotor response was induced by illuminating every eighth LED along the top and bottom of the plate, and progressively shifting the illuminated LED down the strip, resulting in an approximately sinusoidal stimulus, 5.5 cm peak to peak, translating at 5.5 cm/s. Direction of motion was switched every 30 s, for a total testing period of 1 hr, and performance was scored as the average change in heading direction towards the direction of motion during these 30 s epocs. Acoustic tap stimuli were delivered using a Solenoid (ROB-10391, Sparkfun). The behavioural paradigm was designed to be symmetrical such that 1 hr worth of stimulation was followed by 1 hr worth of rest (*Figure 1B*), allowing us to alternate the view of the camera between two plates using 45° incidence hot mirrors (43-958, Edmund Optics) mounted on stepper motors (*Figure 1A*, ROB-09238, Sparkfun), driven by an EasyDriver (ROB-12779, Sparkfun).

Apparatus was controlled using arduino microcontrollers (Teensy 2.0 and 3.2, PJRC) interfaced with custom-written software (Multi-Fish-Tracker), available at https://github.com/haesemeyer/Multi-Tracker, (copy archived at *Haesemeyer, 2023*).

The protocol for assessing behaviour (*Figures 1B and 3B*) consisted of DFs repeated at 1 min intervals, delivered in four training blocks of 60 stimuli, separated by 1 hr of rest (from 0:00 to 8:00, hr:min of the protocol). For analyses, this epoch is separated into periods reflective of the naive response (first five stimuli), and the remaining 235 stimuli during training. From 8:00 to 8:30, no stimuli are delivered and fish are monitored for spontaneous behaviour. From 8:30 to 9:00, fish are given acoustic stimuli, and from 10:00 to 11:00 fish are assayed for the optomotor response and turning towards the direction of motion (light blue). Finally, at 12:00–13:00, larvae are given 60 additional DFs during the test period (red).

## Behavioural analyses

The behaviour of the fish was tracked online at 28 Hz, and 1-s-long videos at 560 Hz were recorded in response to DF and Acoustic Tap stimuli. Offline tracking on recorded videos was performed in MATLAB (MathWorks) using the script 'TrackMultiTrackerTiffStacks_ParallelOnFrames.m', as described previously, to track larval posture (*Randlett et al., 2019*). Tracks were then analysed using Python. Analysis code is available at https://github.com/owenrandlett/lamire_2022, (copy archived at *Randlett, 2023*).

Responses to DFs and taps were identified as movement events that had a bend amplitude greater than 3 rad and 1 rad, respectively. Behavioural fingerprints were created by first calculating the average value for each fish reflecting either the DF response during the specified time period (naive = DFs 1–5, training = DFs 6–240, test = DFs 241–300) or the average response during the entire stimulus period (Acoustic Taps, OMR, Free Swimming). Periods where the tracking data was incomplete were excluded from the analysis. DFs where larvae did not respond were excluded from the behavioural components other than the probability of response. The SSMD was then calculated for each of these average fish values for the compound-treated larvae relative to the vehicle (DMSO) control (*Figure 3C*). The threshold for determining hit compounds was set at $|SSMD| \geq 2$. These analyses were performed using Analyze_MultiTracker_TwoMeasures.py.

Hierarchical clustering (*Figures 3D and 4A–C*) was performed using SciPy (*Virtanen et al., 2020*). Correlations across different behavioural measures (*Figure 4B*) were calculated computing all pairwise comparisons for each behavioural measure in the SSMD fingerprint across the 176 hit compounds.

Further details and code for the analyses used to create the figure panels are provided in the following notebook: 2022_LamireEtAl_BehavFigs.ipynb. Analyses made use of open-source Python packages, including NumPy (*Harris et al., 2020*), SciPy (*Virtanen et al., 2020*), matplotlib (*Hunter, 2007*), seaborn (*Waskom, 2021*), and open-cv (*Bradski, 2000*).

## Pharmacology

Compounds were prepared as 1000× frozen stock solutions in DMSO. Stock solutions were initially diluted 1:100 in E3, yielding a 10× solution. Then, 30 uL of this solution was pipetted into the wells, yielding a 1× compound solution in 0.1% DMSO (Sigma). Vehicle treatment followed the same

protocol using pure DMSO. Larvae were incubated in compound solution for between 30 and 90 min prior to behavioural testing.

The small-molecule compound library (Selleckchem Bioactive: FDA-approved/FDA-like small molecules *Figure 3—source data 1*) was obtained from the UPenn High-Throughput Screening Core. The library concentration was 10 mM, and thus all compounds were screened at approximately 10 µM. For subsequent pharmacological experiments, chemicals were obtained from picrotoxinin: Sigma, P-8390; melatonin: Cayman, 14427; Sigma, M5250; ethinyl estradiol: Cayman, 10006486; and hexestrol: Sigma, H7753.

## Microscopy

Imaging was performed on 5 dpf larvae, mounted tail-free in 2% LMP agarose (Sigma A9414) in E3, using a 20× 1.0 NA water dipping objective (Olympus). Volumetric imaging (*Figures 1, 2, and 6*) was performed at 930 nm on a Bruker Ultima microscope at the CIQLE imaging platform (Lyon, LYMIC) using a resonant scanner resonant scanner over a rectangular region of 1024 × 512 pixels (0.6 µm x/y resolution) and piezo objective mount for fast z-scanning. Imaging sessions began by taking an 'Anatomy Stack' consisting of 150 slices at 1 µm z-steps, summed over 128 repeats (imaging time ≈11 min). This served as the reference stack used for alignment to the Z-Brain atlas and detect Z-drift in the imaging session (see below). The functional stack consisted of 12 slices separated at 10 µm steps, thus covering 120 µm in the brain acquired at 1.98 Hz. To image *Tg(elavl3:H2B-GCaMP6s);Tg(gad1b:DsRed)* double transgenic larvae (*Figure 7*), we used a custom-built two-photon microscope (*Haesemeyer et al., 2018*), imaging 512 × 512 images (0.98 µm x/y resolution) at 1.05 Hz. The anatomy stack was taken at 2 µm step sizes for both the green and red channels in the dark. Functional imaging was performed only on the green/GCaMP channel since the red stimulus LED was incompatible with DsRed imaging.

When developing this protocol, we determined that substantial shifts of more than a cell-body diameter (5 uM) in the Z-plane are common during the ≈1.2 hr of imaging. We determined this by comparing the sum of the functional image planes during five equally sized time epochs (1540 frames per epoch), aligned to the 'Anatomy Stack,' using 'phase_cross_correlation' in the scikit-image library (*van der Walt et al., 2014*). This allowed us to quantify shifts in the imaging plane as shifts in this alignment. These tended to occur within the first hour of imaging; therefore, we performed an hour of imaging of this functional stack before beginning the DF stimulation protocol to allow the preparation to settle under imaging conditions. DFs were delivered using a 3 mm red LED mounted above the fish, controlled by an Arduino Nano connected to the microscope GPIO board and the Prairie View software to deliver pulses of darkness consisting of 1 s light off, 20 s linear ramp back to light on, delivered at 60 s intervals.

Even with this pre-imaging protocol, z-shifts were still observed in a considerable number of fish. Since our habituation-based analysis is focused on how individual neurons change their responses over time, shifts in the z-plane are extremely problematic as they are not correctable post-acquisition and can result in different neurons being imaged at individual voxels. This could easily be confused for changes in functional responses over time during habituation. Therefore, any fish showing a z-drift of greater than 3 µm was excluded from our analysis. Stable z-positioning was further confirmed by manual inspection of the eigen images in the imaging time course using 'View registration metrics' in suite2p to confirm that these do not reflect z-drift. Of the 56 larvae imagined in total, 22 were excluded, leaving 34 included. Larvae were treated with 0.1% DMSO, picrotoxinin (PTX, 10 uM), or melatonin (1 uM) approximately 1 hr before imaging. These fish were analysed as a single population (*Figures 1 and 2*) and separately to determine the effects of the treatments (*Figure 6*).

To quantify responses to the DF stimuli, we used motion artefacts in the imaging data to identify frames associated with movements (*Figure 1—figure supplement 1*). Motion artefact was quantified using the 'corrXY' parameter from suite2p, which reflects the peak of phase correlation comparing each acquired frame and reference image used for motion correction. The 'motion power' was quantified as the standard deviation of a three-frame rolling window, which was smoothed in time using a Savitzky–Golay filter (window length = 15 frames, polyorder = 2). A response to a DF was defined as a 'motion power' signal greater than 3 (z-score) occurring within 10 s of the DF onset and was used to quantify habituation in the head-embedded preparation (*Figure 6A*).

## Ca²⁺ imaging analysis

ROIs were identified using suite2p (*Pachitariu et al., 2017*) using the parameters outlined in RunSuite2p_BrukerData_ScreenPaper.py and RunSuite2p_MartinPhotonData_ScreenPaper.py scripts for the data from the Bruker Ultima microscope (*Figures 5–7*) and custom-built two-photon microscope (*Figure 7F and G*), respectively. These ROIs mostly reflected individual neuronal nuclei/soma. The clustered heatmap image of neural activity (*Figure 3F*) was generated using the suite2p GUI using the 'Visualize selected cells' function and sorting the neurons using the rastermap algorithm (*Pachitariu et al., 2017*, https://github.com/MouseLand/rastermap, copy archived at *MouseLand, 2023*). The imaging planes were then aligned to the anatomical stack taken before functional imaging using 'phase_cross_correlation' in the scikit-image library (*van der Walt et al., 2014*). For the volumetric data, the anatomical stack was then aligned to the Z-Brain atlas coordinates using CMTK, and ROI coordinates were transformed into Z-Brain coordinates using streamxform in CMTK. These steps were performed using Bruker2p_AnalyzePlanesAndRegister.py.

To identify ROIs that were correlated with the stimulus, we used a regression-based approach (*Miri et al., 2011*), where we identified ROIs that were correlated with vectors representing the time course of the DF stimuli convolved with a kernel approximating the slowed H2B-GCaMP time course with respect to neuronal activity. These regressors reflected either the entire 21 s DF stimulus, or only the onset of the flash, and either the first three, last three, or all 60 flashes (six regressors in total). To identify neurons correlated to motor output, we took advantage of the plane-based registration statistics calculated by suite2p. Specifically, the 'ops['corrXY']' metric, which reflects the correlation of each registered image frame with the reference image. We reasoned that movements would cause image artefacts and distortions that would be reflected as a transient drop in these correlations. Indeed, we confirmed this association by imaging the tail using an infrared camera and compared the motion index calculated through tail tracking, and that which we calculated based on the motion artefacts, which showed good overall agreement in predicted movement events and average correlation of 0.4, demonstrating that these image-based artefacts can be used as reliable proxies of tail movements (*Figure 1—figure supplement 1*). Therefore, regressors based on these motion indices were used to identify neurons correlated with motor output.

Images for the functional tuning of individual neurons (*Figure 1G–J*) were computed using the hue saturation value (HSV) colour scheme, with the maximal correlation value to either regressor mapped to saturation, and the hue value reflecting the linear preference for either regressor. Clustering of functional response types (*Figure 2*) was done by first selecting all those ROIs that showed a correlation of ≥0.25 with any of the six stimulus regressors across all imaged fish. Then among these ROIs we removed any ROIs that did not show a correlation of ≥0.3 with at least five ROIs imaged in a different larvae. This filtered out ROIs that were unique in any individual fish, allowing us to focus on those neuron types that were most consistent across individuals. We then used the Affinity Propagation clustering from scikit-learn (*Pedregosa et al., 2011*), with 'affinity' computed as the Pearson product–moment correlation coefficients (corrcoef in NumPy; *Harris et al., 2020*), preference = -9 and damping = 0.9, and clustered using hierarchical clustering (cluster.hierarchy in SciPy; *Virtanen et al., 2020*). Cluster number was assigned based on the ordering of the hierarchical clustering tree.

To generate the final cluster assignments, we re-scanned all the ROIs calculating their correlation with the mean-response vectors for each of the identified 12 functional clusters, selecting those with a correlation value of ≥0.3, which were then assigned to the cluster with which they had the highest correlation. To determine the cluster assignments for the data from *Tg(Gad1b:DsRed);Tg(elavl3:H2B-GCaMP6s)* double transgenic larvae (*Figure 7F and G*), data were realigned and interpolated to match the frame rate of the clustered data, and assigned to the 12 clusters as above.

To compare the spatial relationships between the neuronal positions of different functional clusters (*Figure 2E*), and between the functional clusters and reference brain labels (*Figure 7A–E*), image volumes were cropped to the imaged coordinates (*Figure 1E*), downsampled to isometric 10 um³ voxels, and linearized to calculate the Pearson correlation coefficient between the image sub-volumes.

Analyses made use of multiple open-source Python packages, including suite2p (*Pachitariu et al., 2017*), NumPy (*Harris et al., 2020*), SciPy (*Virtanen et al., 2020*), scikit-learn (*Pedregosa et al., 2011*), scikit-image (*van der Walt et al., 2014*), numba (*Lam et al., 2015*), matplotlib (*Hunter, 2007*), seaborn (*Waskom, 2021*), and open-cv (*Bradski, 2000*). Details of the analyses used to create the figure panels are provided in the following notebook: 2022_LamireEtAl_FunctionalFigs.ipynb.

## Acknowledgements

We thank the Randlett, Granato, and Engert group members for helpful advice regarding the manuscript and work, Gregory Forkin for help in zebrafish husbandry, the Burgess and Baier groups for sharing anatomical data from the Zebrafish Brain Browser and mapZebrain atlases, and Armin Bahl for sharing the mapZebrain to Z-Brain transformation. This work was supported by funding from the ATIP-Avenir program of the CNRS and Inserm (OR), a Fondation Fyssen research grant (OR), the IDEX-Impulsion initiative of the University of Lyon (OR), the NIH grants MH109498 (MG), NS123887 (MH), U19NS104653 and 1R01NS124017 (FE), the National Science Foundation grant IIS-1912293 (FE), and the Simons Foundation grant SCGB 542973 (FE).

## Additional information

### Funding

| Funder | Grant reference number | Author |
| --- | --- | --- |
| Institut National de la Santé et de la Recherche Médicale | ATIP-Avenir | Owen Randlett |
| Fondation Fyssen | Research Grant | Owen Randlett |
| Université de Lyon | IDEX-Impulsion | Owen Randlett |
| National Institutes of Health | MH109498 | Michael Granato |
| National Institutes of Health | NS123887 | Martin Haesemeyer |
| National Institutes of Health | U19NS104653 | Florian Engert |
| National Science Foundation | IIS- 1912293 | Florian Engert |
| Simons Foundation | SCGB 542973 | Florian Engert |
| National Institutes of Health | 1R01NS124017 | Florian Engert |

The funders had no role in study design, data collection and interpretation, or the decision to submit the work for publication.

### Author contributions

Laurie Anne Lamiré, Investigation, Writing – review and editing; Martin Haesemeyer, Software, Writing – review and editing; Florian Engert, Resources, Supervision, Funding acquisition, Writing – review and editing; Michael Granato, Conceptualization, Resources, Supervision, Funding acquisition, Writing – review and editing; Owen Randlett, Conceptualization, Resources, Data curation, Software, Formal analysis, Supervision, Funding acquisition, Validation, Investigation, Visualization, Methodology, Writing - original draft, Writing – review and editing

### Author ORCIDs

Martin Haesemeyer http://orcid.org/0000-0003-2704-3601
Florian Engert https://orcid.org/0000-0001-8169-2990
Owen Randlett http://orcid.org/0000-0003-0181-5239

### Ethics

Zebrafish were housed, cared for, and bred at the Harvard MCB, UPenn CDB, and Lyon PRECI zebrafish facilities. All experiments were done in accordance with relevant approval from local ethical committees at Harvard University, the University of Pennsylvania, and the University of Lyon.

Reviewer #1 (Public Review): https://doi.org/10.7554/eLife.84926.3.sa1

Reviewer #2 (Public Review): https://doi.org/10.7554/eLife.84926.3.sa2
Reviewer #3 (Public Review): https://doi.org/10.7554/eLife.84926.3.sa3
Author Response https://doi.org/10.7554/eLife.84926.3.sa4

## Additional files

### Supplementary files
• MDAR checklist

### Data availability
Code for data analysis and for generating the figure panels is available here: https://github.com/owen-randlett/lamire_2022 (copy archived at *Randlett, 2023*). Data are available here: https://lamire2022.randlettlab.com/ or here: https://doi.org/10.5061/dryad.jdfn2z3fc.

The following dataset was generated:

| Author(s) | Year | Dataset title | Dataset URL | Database and Identifier |
|---|---|---|---|---|
| Lamiré L, Haesemeyer M, Engert F, Granato M, Randlett O | 2023 | Data for: Inhibition drives habituation of a larval zebrafish visual response | https://doi.org/10.5061/dryad.jdfn2z3fc | Dryad Digital Repository, 10.5061/dryad.jdfn2z3fc |

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
