## [Editor Report · eLife assessment]

This **valuable** manuscript attempts to identify the brain regions and cell types involved in habituation to dark flash stimuli in larval zebrafish. Habituation being a form of learning widespread in the animal kingdom, the investigation of neural mechanisms underlying it is a worthwhile endeavor. The authors use a combination of behavioral analysis, neural activity imaging, and pharmacological manipulation to investigate brain-wide mechanisms of habituation. While the data presented are **solid**, the authors conclude that there is no simple relationship between pharmacological intervention, neural activity patterns, and behavioral outcomes, and a robust causative link can therefore not be established.

---

## [Referee Report · Reviewer #1 (Public Review)]

This manuscript addresses the important and understudied issue of circuit-level mechanisms supporting habituation, particularly in pursuit of the possible role of increases in the activity of inhibitory neurons in suppressing behavioral output during long-term habituation. The authors make use of many of the striking advantages of the larval zebrafish to perform whole brain, single neuronal calcium imaging during repeated sensory exposure, and high throughput screening of pharmacological agents in freely moving, habituating larvae. Notably, several blockers/antagonists of GABAA(C) receptors completely suppress habituation of the O-bend escape response to dark flashes, suggesting a key role for GABAergic transmission in this form of habituation. Other substances are identified that strikingly enhance habituation, including melatonin, although here the suggested mechanistic insight is less specific. To add to these findings, a number of functional clusters of neurons are identified in the larval brain that have divergent activity through habituation, with many clusters exhibiting suppression of different degrees, in line with adaptive filtration during habituation, and a single cluster that potentiates during habituation. Further assessment reveals that all of these clusters include GABAergic inhibitory neurons and excitatory neurons, so we cannot take away the simple interpretation that the potentiating cluster of neurons is inhibitory and therefore exerts an influence on the other adapting (depressing) clusters to produce habituation. Rather, a variety of interpretations remain in play.

Overall, there is great potential in the approach that has been used here to gain insight into circuit-level mechanisms of habituation. There are many experiments performed by the authors that cannot be achieved currently in other vertebrate systems, so the manuscript serves as a potential methodological platform that can be used to support a rich array of future work. While there are several key observations that one can take away from this manuscript, a clear interpretation of the role of GABAergic inhibitory neurons in habituation has not been established. This potential feature of habituation is emphasized throughout, particularly in the introduction and discussion sections, meaning that one is obliged as a reader to interrogate whether the results as they currently stand really do demonstrate a role for GABAergic inhibition in habituation. Currently, the key piece of evidence that may support this conclusion is that picrotoxin, which acts to block some classes of GABA receptors, prevents habituation. However, there are interpretations of this finding that do not specifically require a role for modified GABAergic inhibition. For instance, by lowering GABAergic inhibition, an overall increase in neural activity will occur within the brain, in this case below a level that could cause a seizure. That increase in activity may simply prevent learning by massively increasing neural noise and therefore either preventing synaptic plasticity or, more likely, causing indiscriminate synaptic strengthening and weakening that occludes information storage. Sensory processing itself could also be disrupted, for instance by altering the selectivity of receptive fields. Alternatively, it could be that the increase in neural activity produced by the blockade of inhibition simply drives more behavioral output, meaning that more excitatory synaptic adaptation is required to suppress that output. The authors propose two specific working models of the ways in which GABAergic inhibition could be implemented in habituation. An alternative model, in which GABAergic neurons are not themselves modified but act as a key intermediary between Hebbian assemblies of excitatory neurons that are modified to support memory and output neurons, is not explored. As yet, these or other models in which inhibition is not required for habituation, have not been fully tested.

This manuscript describes a really substantial body of work that provides evidence of functional clusters of neurons with divergent responses to repeated sensory input and an array of pharmacological agents that can influence the rate of a fundamentally important form of learning.

---

## [Referee Report · Reviewer #2 (Public Review)]

In this study, Lamire et al. use a calcium imaging approach, behavioural tests, and pharmacological manipulations to identify the molecular mechanisms behind visual habituation. They show a valuable drug screen paradigm to assess the impact of pharmacological compounds on the behaviour of larval zebrafish.

The pharmacological screen identifies an expected suppression of habituation by GABA receptor antagonists. More interestingly, it identifies potentially new contributions of melatonin receptor agonists, and oestrogen receptor agonists to habituation, as they seem to increase the rate of habituation.

The volumetric calcium imaging of habituation to dark flashes is valuable, but the mix of responses to visual cues that are not relevant to the dark flash escape, such as the slow increase back to baseline luminosity, lowers the clarity of the results. The link between the calcium imaging results and free-swimming behaviour is not especially convincing, however, that is a common issue of head-restrained imaging with larval zebrafish. The identification of a cluster of neurons with potentiating responses, which could drive the habituation is intriguing, but more characterizations of these neurons would be needed to fully understand their function in habituation. The pharmacological manipulation of the habituation circuits mapped in the first part does not arrive at any satisfying conclusion, which is acknowledged by the authors.

Overall, the authors did identify interesting new molecular pathways that may be involved in habituation to dark flashes. Their screening approach, while not novel, will be a powerful way to interrogate other behavioural profiles. The authors identified circuit loci apparently involved in habituation to dark flashes, and the potentiation and no adaptation clusters have not been previously observed and are interesting targets for future work. This work suggests that the circuits and mechanisms underlying habituation are likely more complex than anticipated. The data will be useful to guide follow-up experiments by the community on the new pathway candidates that this screen has uncovered, including behaviours beyond dark flash habituation.

---

## [Referee Report · Reviewer #3 (Public Review)]

To analyze the circuit mechanisms leading to the habituation of the O-bed responses upon repeated dark flashes (DFs), the authors performed 2-photon Ca2+ imaging in larvae expressing nuclear-targeted GCaMP7f pan-neuronally panning the majority of the midbrain, hindbrain, pretectum, and thalamus. They found that while the majority of neurons across the brain depress their responsiveness during habituation, a smaller population of neurons in the dorsal regions of the brain, including the torus longitudinalis, cerebellum, and dorsal hindbrain, showed the opposite pattern, suggesting that motor-related brain regions contain non-depressed signals, and therefore likely contribute to habituation plasticity.

Further analysis using affinity propagation clustering identified 12 clusters that differed both in their adaptation to repeated DFs, as well as the shape of their response to the DF.

Next by the pharmacological screening of 1953 small molecule compounds with known targets in conjunction with the high-throughput assay, they found that 176 compounds significantly altered some aspects of measured behavior. Among them, they sought to identify the compounds that (1) have minimal effects on the naive response to DFs, but strong effects during the training and/or memory retention periods, (2) have minimal effects on other aspects of behaviors, (3) show similar behavioral effects to other compounds tested in the same molecular pathway, and identified the GABAA/C Receptor antagonists Bicuculline, Amoxapine, and Picrotoxinin (PTX). As partial antagonism of GABAAR and/or GABACR is sufficient to strongly suppress habituation but not generalized behavioral excitability, they concluded that GABA plays a very prominent role in habituation. They also identified multiple agonists of both Melatonin and Estrogen receptors, indicating that hormonal signalling may also play a prominent role in habituation response.

To integrate the results of the Ca2+ imaging experiments with the pharmacological screening results, the authors compared the Ca2+ activity patterns after treatment with vehicle, PTX, or Melatonin in the tethered larvae. The behavioral effects of PTX and Melatonin were much smaller compared with the very strong behavioral effects in freely-swimming animals, but the authors assumed that the difference was significant enough to continue further experiments. Based on the hypothesis that Melatonin and GABA cooperate during habituation, they expected PTX and Melatonin to have opposite effects. This was not the case in their results: for example, the size of the 12(Pot, M) neuron population was increased by both PTX and Melatonin, suggesting that pharmacological manipulations that affect habituation behavior manifest in complex functional alterations in the circuit, making capturing these effects by a simple difficult.

Since the 12(*Pot, M*) neurons potentiate their responses and thus could act to progressively depress the responses of other neuronal classes, they examined the identity of these neurons with GABA neurons. However, GABAergic neurons in the habituating circuit are not characterized by their Adaptation Profile, suggesting that global manipulations of GABAergic signalling through PTX have complex manifestations in the functional properties of neurons.

Overall, the authors have performed an admirably large amount of work both in whole-brain neural activity imaging and pharmacological screening.

---

## [Author Response]

The following is the authors’ response to the original reviews.

**eLife assessment**
This valuable manuscript attempts to identify the brain regions and cell types involved in habituation to dark flash stimuli in larval zebrafish. Habituation being a form of learning widespread in the animal kingdom, the investigation of neural mechanisms underlying it is an important endeavor. The authors use a combination of behavioral analysis, neural activity imaging, and pharmacological manipulation to investigate brain-wide mechanisms of habituation. However, the data presented are incomplete and do not show a convincing causative link between pharmacological manipulations, neural activity patterns, and behavioral outcomes.

We thank the reviewers and editors for their careful reading and reviews of our work. We are grateful that they appreciate the value in our experimental approach and results. We acknowledge what we interpret as the major criticism, that in our original manuscript we focused too heavily on the hypothesized role of GABAergic neurons in driving habituation. This hypothesis will remain only indirectly supported until we can identify a GABAergic population of neurons that drives habituation. Therefore, we have revised our manuscript, decreasing the focus on GABA, and rather emphasizing the following three points:

1. By performing the first Ca2+ imaging experiments during dark flash habituation, we identify multiple distinct functional classes of neurons which have different adaptation profiles, including non-adapting and potentiating classes. These neurons are spread throughout the brain, indicating that habituation is a complex and distributed process.

2. By performing a pharmacological screen for dark flash habituation modifiers, we confirm habituation behaviour manifests from multiple distinct molecular mechanisms that independently modulate different behavioural outputs. We also implicate multiple novel pathways in habituation plasticity, some of which we have validated through dose-response studies.

3. By combining pharmacology and Ca2+ imaging, we did not observe a simple relationship between the behavioural effects of a drug treatment and functional alterations in neurons. This observation further supports our model that habituation is a multidimensional process, for which a simple circuit model will be insufficient.

We would like to point out that, in our opinion, there appears to be a factual error in the final sentence of the eLife assessment:

“However, the data presented are incomplete and do not show a convincing causative link betweenpharmacological manipulations, neural activity patterns, and behavioral outcomes”.

We believe that a “convincing causative link” between pharmacological manipulations and behavioural outcomes has been clearly demonstrated for PTX, Melatonin, Estradiol and Hexestrol through our dose response experiments. Similarly a link between pharmacology and neural activity patterns has also been directly demonstrated. As mentioned in (3), we acknowledge that our data linking neural activity and behaviour is more tenuous, as will be more explicitly reflected in our revised manuscript.

Nevertheless, we maintain that one of the primary strengths of our study is our attempt to integrate analyses that span the behavioural, pharmacological, and neural activity-levels.

In our revised manuscript, we have substantially altered the Abstract and Discussion, removed the Model figure (previously Figure 8), and changed the title from :

“Inhibition drives habituation of a larval zebrafish visual response”

to:

“Functional and pharmacological analyses of visual habituation learning in larval zebrafish”

Text changes from the initial version are visible as track changes in the word document: “LamireEtAl_2022_eLifeRevisions.docx”

**Reviewer #1 (Public Review):**
This manuscript addresses the important and understudied issue of circuit-level mechanisms supporting habituation, particularly in pursuit of the possible role of increases in the activity of inhibitory neurons in suppressing behavioral output during long-term habituation. The authors make use of many of the striking advantages of the larval zebrafish to perform whole brain, single neuronal calcium imaging during repeated sensory exposure, and high throughput screening of pharmacological agents in freely moving, habituating larvae. Notably, several blockers/antagonists of GABAA(C) receptors completely suppress habituation of the O-bend escape response to dark flashes, suggesting a key role for GABAergic transmission in this form of habituation. Other substances are identified that strikingly enhance habituation, including melatonin, although here the suggested mechanistic insight is less specific. To add to these findings, a number of functional clusters of neurons are identified in the larval brain that has divergent activity through habituation, with many clusters exhibiting suppression of different degrees, in line with adaptive filtration during habituation, and a single cluster that potentiates during habituation. Further assessment reveals that all of these clusters include GABAergic inhibitory neurons and excitatory neurons, so we cannot take away the simple interpretation that the potentiating cluster of neurons is inhibitory and therefore exerts an influence on the other adapting (depressing) clusters to produce habituation. Rather, a variety of interpretations remain in play.Overall, there is great potential in the approach that has been used here to gain insight intocircuit-level mechanisms of habituation. There are many experiments performed by the authors that cannot be achieved currently in other vertebrate systems, so the manuscript serves as a potential methodological platform that can be used to support a rich array of future work. While there are several key observations that one can take away from this manuscript, a clear interpretation of the role of GABAergic inhibitory neurons in habituation has not been established. This potential feature of habituation is emphasized throughout, particularly in the introduction and discussion sections, meaning that one is obliged as a reader to interrogate whether the results as they currently stand really do demonstrate a role for GABAergic inhibition in habituation. Currently, the key piece of evidence that may support this conclusion is that picrotoxin, which acts to block some classes of GABA receptors, prevents habituation. However, there are interpretations of this finding that do not specifically require a role for modified GABAergic inhibition. For instance, by lowering GABAergic inhibition, an overall increase in neural activity will occur within the brain, in this case below a level that could cause a seizure. That increase in activity may simply prevent learning by massively increasing neural noise and therefore either preventing synaptic plasticity or, more likely, causing indiscriminate synaptic strengthening and weakening that occludes information storage. Sensory processing itself could also be disrupted, for instance by altering the selectivity of receptive fields. Alternatively, it could be that the increase in neural activity produced by the blockade of inhibition simply drives more behavioral output, meaning that more excitatory synaptic adaptation is required to suppress that output. The authors propose two specific working models of the ways in which GABAergic inhibition could be implemented in habituation. An alternative model, in which GABAergic neurons are not themselves modified but act as a key intermediary between Hebbian assemblies of excitatory neurons that are modified to support memory and output neurons, is not explored. As yet, these or other models in which inhibition is not required for habituation, have not been fully tested.This manuscript describes a really substantial body of work that provides evidence of functional clusters of neurons with divergent responses to repeated sensory input and an array of pharmacological agents that can influence the rate of a fundamentally important form of learning.

We thank the reviewer for their careful consideration of our work, and we agree that multiple models of how habituation occurs remain plausible. As discussed above and below in more detail, we have revised our manuscript to better reflect this. We hope the reviewer will agree that this has improved the manuscript.

**Reviewer #2 (Public Review):**
In this study, Lamire et al. use a calcium imaging approach, behavioural tests, and pharmacological manipulations to identify the molecular mechanisms behind visual habituation. Overall, the manuscript is well-written but difficult to follow at times. They show a valuable new drug screen paradigm to assess the impact of pharmacological compounds on the behaviour of larval zebrafish, the results are convincing, but the description of the work is sometimes confusing and lacking details.

We thank the reviewer for identifying areas where our description lacked details. We apologize for these omissions and have attempted to add relevant details as described below. We note that all of the analysis code is available online, though we appreciate that navigating and extracting data from these files is not straightforward.

The volumetric calcium imaging of habituation to dark flashes is valuable, but the mix of responses to visual cues that are not relevant to the dark flash escape, such as the slow increase back to baseline luminosity, lowers the clarity of the results. The link between the calcium imaging results and free-swimming behaviour is not especially convincing, however, that is a common issue of head-restrained imaging with larval zebrafish.

We agree with the reviewer that the design of our stimulus, and specifically the slow increase back to baseline luminosity, is perhaps confusing for the interpretation of some of the response profiles of neurons. We originally chose this stimulus type (rather than a square wave of 1s of darkness, for example) in order to better highlight the responses of the larvae to the onset of darkness (rather than the response to abruptly returning to full brightness). We therefore believe that the slow return to baseline is an important feature of the stimulus,, which better separates activity related to the fast offset from activity related to light onset. And since all of the foundational behavioural data (Randlett et al., Current Biology 2019), and pharmacological data, used this stimulus type, we did not change it for the Ca2+ imaging experiments. Our use of relatively slow nuclear-targeted GCaMP indicators also means that the temporal resolution of our imaging experiments is relatively poor, and therefore we felt that using a stimulus that highlighted light offset might be best.

We also fully acknowledge in the Results section that the behaviour of the head embedded fish is not the same as that of free-swimming fish, and that therefore establishing a direct link between these types of experiments is complicated. This is an unavoidable caveat in the head-embedded style experiments. To further emphasize this, we have also added a paragraph to the discussion where this is acknowledged explicitly.

“We also found that the same pharmacological treatments that result in strong alterations to habituation behaviour in freely swimming larvae ([fig:5]), resulted in relatively subtle and complex functional alterations in the circuit ([fig:6]). Making direct comparisons between freely-swimming behaviour and head-fixed Ca2+ imaging is always challenging due to the differences in behaviour observed in the two contexts, and therefore our failure to identify a clear logic in these experiments may have technical explanations that will require approaches to measure neural activity from unrestrained and freely-behaving animals to resolve . Alternatively, these results are again consistent with the idea that habituation is a multidimensional and perhaps highly non-linear phenomenon in the circuit, which cannot be captured by a simple model.”

The strong focus on GABA seems unwarranted based on the pharmacological results, as only Picrotoxinin gives clear results, but the other antagonists do not give a consistent results. On the other hand, the melatonin receptor agonists, and oestrogen receptor agonists give more consistent results, including more convincing dose effects.

We agree that our manuscript focused too strongly on GABA and have toned this down. We are currently performing genetic experiments aimed at identifying the Melatonin, Estrogen and GABA receptors that function during habituation, which we think will be necessary to move beyond pharmacology and the necessary caveats that such experiments bring.

The pharmacological manipulation of the habituation circuits mapped in the first part does not arrive at any satisfying conclusion, which is acknowledged by the authors. These results do reinforce the disconnect between the calcium imaging and the behavioural experiments and undercut somewhat the proposed circuit-level model.

We agree with this criticism and have toned down the focus on GABA specifically in the circuit, and have removed the speculative model previously in Figure 8.

Overall, the authors did identify interesting new molecular pathways that may be involved in habituation to dark flashes. Their screening approach, while not novel, will be a powerful way to interrogate other behavioural profiles. The authors identified circuit loci apparently involved in habituation to dark flashes, and the potentiation and no adaptation clusters have not been previously observed as far as I know.The data will be useful to guide follow-up experiments by the community on the new pathway candidates that this screen has uncovered, including behaviours beyond dark flash habituation.

We again thank the reviewer for both their support of our approach, and in pointing out where our conclusions were not well supported by our data.

**Reviewer #3 (Public Review):**
To analyze the circuit mechanisms leading to the habituation of the O-bed responses upon repeated dark flashes (DFs), the authors performed 2-photon Ca2+ imaging in larvae expressingnuclear-targeted GCaMP7f pan-neuronally panning the majority of the midbrain, hindbrain, pretectum, and thalamus. They found that while the majority of neurons across the brain depress their responsiveness during habituation, a smaller population of neurons in the dorsal regions of the brain, including the torus longitudinalis, cerebellum, and dorsal hindbrain, showed the opposite pattern, suggesting that motor-related brain regions contain non-depressed signals, and therefore likely contribute to habituation plasticity.Further analysis using affinity propagation clustering identified 12 clusters that differed both in their adaptation to repeated DFs, as well as the shape of their response to the DF.Next by the pharmacological screening of 1953 small molecule compounds with known targets in conjunction with the high-throughput assay, they found that 176 compounds significantly altered some aspects of measured behavior. Among them, they sought to identify the compounds that (1) have minimal effects on the naive response to DFs, but strong effects during the training and/or memory retention periods, (2) have minimal effects on other aspects of behaviors, (3) show similar behavioral effects to other compounds tested in the same molecular pathway, and identified the GABAA/C Receptor antagonists Bicuculline, Amoxapine, and Picrotoxinin (PTX). As partial antagonism of GABAAR and/or GABACR is sufficient to strongly suppress habituation but not generalized behavioral excitability, they concluded that GABA plays a very prominent role in habituation. They also identified multiple agonists of both Melatonin and Estrogen receptors, indicating that hormonal signaling may also play a prominent role in habituation response.To integrate the results of the Ca2+ imaging experiments with the pharmacological screening results, the authors compared the Ca2+ activity patterns after treatment with vehicle, PTX, or Melatonin in the tethered larvae. The behavioral effects of PTX and Melatonin were much smaller compared with the very strong behavioral effects in freely-swimming animals, but the authors assumed that the difference was significant enough to continue further experiments. Based on the hypothesis that Melatonin and GABA cooperate during habituation, they expected PTX and Melatonin to have opposite effects. This was not the case in their results: for example, the size of the 12(Pot, M) neuron population was increased by both PTX and Melatonin, suggesting that pharmacological manipulations that affect habituation behavior manifest in complex functional alterations in the circuit, making capturing these effects by a simple difficult.Since the 12(*Pot, M*�) neurons potentiate their responses and thus could act to progressively depress the responses of other neuronal classes, they examined the identity of these neurons with GABA neurons. However, GABAergic neurons in the habituating circuit are not characterized by their Adaptation Profile, suggesting that global manipulations of GABAergic signaling through PTX have complex manifestations in the functional properties of neurons.Overall, the authors have performed an admirably large amount of work both in whole-brain neural activity imaging and pharmacological screening. However, they are not successful in integrating the results of both experiments into an acceptably consistent interpretation due to the incongruency of the results of different experiments. Although the authors present some models for interpretation, it is not easy for me to believe that this model would help the readers of this journal to deepen the understanding of the mechanisms for habituation in DF responses at the neural circuit level.This reviewer would rather recommend the authors divide this manuscript into two and publish two papers by adding some more strengthening data for each part such as cellular manipulations, e.g. ablation to prove the critical involvement of 12(Pot, M) neurons in habituation.

We thank the reviewer for their careful consideration of our manuscript, and we agree that our emphasis on a particular model of DF habituation, namely the potentiation of GABAergic synapses, was overly speculative. We hope they will agree that our revised manuscript better reflect the results from our experiments, and we have tried to more specifically emphasize the incongruency in our behavioural and Ca2+ imaging data after pharmacological treatment, which we agree shows that a simple model is insufficient to capture both of these sets of observations.

We have opted not to split the paper into two, since we feel that the collective message of this paper and approach combining molecular and functional analysis will be of interest. Moreover, we feel that the molecular and functional analyses feed off of each other and provide a level of complementarity that would be lost if the manuscript would be split, even if the message in this particular case is rather complex

**Reviewer #1 (Recommendations For The Authors):**
There is much to commend about this manuscript. The advantages of studying habituation in the zebrafish larva are very clearly demonstrated, including the wonderful calcium imaging across the brain and the relatively high throughput screening of large numbers of different pharmacological agents. The habituation to dark flashes in freely moving larvae is also striking and the very large effect size serves the screening beautifully. Thus, if we take the really substantial amount of work of a very high standard that has been done here, there is clearly potential for an important new contribution to the literature. However, as you will see from my public review, I am of the opinion that a specific role for the modification of GABAergic inhibitory systems has not yet been established through this work. While the potential role for GABAergic inhibitory neurons in habituation, either as the key modifiable element or as an intermediary between memory and motor output, is an attractive theory with many strengths, your study as it currently stands does not categorically demonstrate that one of those two options holds. For instance, the more traditional view, that adaptive filtration is mediated by weakened synaptic connectivity between excitatory sensory systems and excitatory motor output or reduced intrinsic excitability in those same neurons, could still be in operation here. By lowering GABAergic influence over post-synaptic targets with picrotoxin, it is possible that motor output remains highly active, and even lower activity or synaptic drive from those excitatory sensory systems that feed into the output may still reliably produce behavioral output. Alternatively, it could be the formation of a memory of the familiar stimulus is disrupted by reduced inhibition that alters sensory coding either by introducing noise or reducing the selectivity of receptive fields. I believe that there are several options to address these concerns:1. You could change the emphasis of the manuscript so that it is less focused on inhibition and instead emphasizes the categorization of clusters of neurons that have divergent responses during habituation, including either strong suppression to potentiation. To this, you add a high throughput screening system with a wide range of different agents being tested, several of which produce a significant effect on habituation in either direction. These observations in themselves provide powerful building blocks for future work.1. If GABAergic neurons play a key role in habituation in this paradigm, then picrotoxin is having its effect by blocking receptors on excitatory neurons. Thus, it seems that selectively imaging GABAergic neurons before and after the application of these drugs is not likely to reveal the contribution of GABAergic synaptic influence on excitatory targets. More important is to get a stronger sense of how the GABAergic neurons change their activity throughout habituation and then influence the downstream target neurons of those GABAergic neurons (some of which may themselves be inhibitory and participating in disinhibition). For instance, you could interrogate whether anti-correlations in activity levels exist between presynaptic inhibitory neurons and putative post-synaptic targets. This analysis could be further bolstered by removing that relationship in the presence of Picrotoxin, thereby demonstrating a direct influence of inhibition from a GABAergic presynaptic partner on a postsynaptic target. While this would constitute a lot more work, it is likely to yield greater insight into a specific role for GABAergic neurons in habituation, and I suspect much of that information is in the existing datasets.1. To really reveal causal roles for inhibition in this form of habituation, it seems to me that there needs to be some selective intervention in GABAergic neuronal activity, ideally bidirectionally, to transiently interrupt or enhance habituation. Optogenetic or chemogenetic stimulation/inactivation is one option in this regard, which I imagine would be challenging to implement and certainly involves a lot of further work, particularly if you are then going to target specific subpopulations of GABAergic neurons. I appreciate that this option seems way beyond the scope of a review process and would probably constitute a follow-up study.

We agree with the reviewer that we have not “categorically demonstrated” that GABAergic inhibitory neurons drive habituation by increasing their influence on the circuit, and appreciate the suggestions for how to reformulate our manuscript to better reflect this. We have opted to follow suggestion (1), and have considerably changed the focus of the manuscript.

The additional analysis suggested in (2) is very interesting, but since we can not identify which cells are inhibitory in our imaging experiments with picrotoxinin treatment, nor which are pre- orpost-synaptic, we feel that this analysis will be very unconstrained. Also, if GABA is acting as an inhibitory neurotransmitter, it therefore is expected to act to drive anticorrelations among pre and postsynaptic neurons through inhibition. Therefore, blockage of GABA through PTX would be expected to result in increased correlations, regardless of our hypothesized role of neurons during habituation. Our current efforts are aimed at identifying critical neurons driving habituation plasticity, and we will perform such analysis once we have mechanisms for identifying these neurons.

Finally, we agree that (3) is the obvious and only way to demonstrate causation here, and this is where we are working towards. However, since we currently have no means of genetically targeting these neurons, we are not able to perform these suggested experiments today.

I have some additional concerns that I would really appreciate you addressing:1. The behavioral habituation is striking in the freely moving larvae, but very hard to monitor in the larvae that are immobilized for calcium imaging. Are there steps that could be taken in the long run to improve direct observation of the habituation effect in these semi-stationary fish? For instance, is it possible to observe eye movements or some more subtle behavioral readout than the O-bend reflex? I apologize if this is a naïve question, but I am not entirely familiar with this specific experimental paradigm.

In the Dark Flash paradigm, we do not have readouts beyond the “O-bend” response itself, which is characterized by a large-angle bend of the tail and turning maneuver. We have not observed other, more subtle behavioural responses, such as eye or fin movements, for example. If we would be able to identify alternative behavioural outputs that were more robustly performed during head-embedded preparations, this would indeed be an advantage allowing us to more directly interpret the Ca2+ imaging results with respect to behaviour.

1. The dark flash as a stimulus to which the larvae habituate is obviously used as a powerful and ethologically relevant stimulus. However, it does leave an element of traditional habituation paradigms out, which is a novel stimulus that can be used to immediately re-instate the habituated response (otherwise known as dishabituation). Is there a way that you can imagine implementing that with zebrafish larvae, for instance through systematically altering a visual feature, such as spatial frequency or orientation? This would be a powerful development in my view as it would not only allow you to rule out motor or sensory fatigue as an underlying cause of reduced behavior but also it would provide an extra feature that strengthens your assessment of neuronal response profiles in candidate populations of inhibitory and excitatory neurons.

We agree that identifying a dishabituating stimulus would be very powerful for our experiments. For short-term habituation of the acoustic startle response, Wolman et al demonstrated that dishabituation occurs after a touch stimulus (Wolman et al., PNAS, 2011; https://doi.org/10.1073/pnas.1107156108). We attempted to dishabituate the O-Bend response with tap and touch stimuli, and this unfortunately did not occur. Our understanding of dishabituation is that this generally requires a second stimulus that elicits the same behaviour as the habituated stimulus (e.g. both acoustic and touch-stimuli elicit the Mauthner-dependent C-bend response). In zebrafish the only stimulus that has been identified that elicits the O-bend is a dark-flash. This lack of an appropriate alternative stimulus is perhaps why we have been unsuccessful in identifying a dishabituating stimulus.

1. You have written about the concept of 'short' and 'long' response shapes when using calcium imaging as a proxy for neural activity, surmising that the short response shape may reflect transient bursting. Although calcium imaging obviously has many advantages, this feature reveals one notable limitation of calcium imaging in contrast to electrophysiology, in that the time course of the signal is considerably longer and does not allow you with confidence to fully detect the response profile of neurons. Is there some kind of further deconvolution process that you could implement to improve the fidelity of your calcium imaging to the occurrence of action potentials? The burstiness of neurons is obviously important as it can indicate a particular type of neuron (for instance fast-spiking inhibitory neurons) or it might reveal a changing influence on post-synaptic neurons. For instance, bursting can be a response to inhibition due to the triggering of T-type calcium channels in response to hyperpolarization.

One of the major limitations to Ca2+ imaging is the lack of temporal resolution. In our particular approach, using nuclear-targeted H2B-GCaMP indicators, further reduces our temporal resolution.Deconvolution approaches can be used in some instances to approximate spike rate, since therise-time of Ca2+ indicators can be relatively fast. However, in our imaging we chose to image larger volumes at the expense of scan rate, where our imaging is performed at only 2hz. Therefore, deconvolution and spike-rate estimation is not appropriate. Considering these limitations, we would argue that the fact that we can observe differences in kinetics of the 'short' and 'long' response shapes indicates that they likely show very different response kinetics, which we hope to confirm by electrophysiology once we have established ways of targeting these neurons for recordings.

1. I note that among the many substances you screened with is MK801. An obvious candidate mechanism in habituation is the NMDA receptor, given the importance of this receptor for so many forms of learning and bidirectional synaptic plasticity. If I am to understand correctly, this NMDA receptor blocker actually enhances habituation in the zebrafish larvae, similar to melatonin. That is a very surprising observation, which is worth looking into further or at least discussed in the manuscript. The finding would, at least, be consistent with the idea that plasticity is not occurring at excitatory synapses and could potentially bolster the argument that plasticity of inhibitory synapses is at play in this particular form of habituation.

This is a very important point. We were also particularly interested in MK801, which has been shown to inhibit other forms of habituation, like short-term acoustic habituation (Wolman et al., PNAS, 2011; https://doi.org/10.1073/pnas.1107156108). In our experiments we did see that fish become even less responsive to dark flashes when treated with MK-801 (SSMD fingerprint data: Prob-Train = -0.39, Prob-Test = -1.58) which would indicate that MK-801 promotes dark flash habituation, similar to Melatonin. However, we also observed that MK-801 caused a decrease in the performance in the other visual assay we tested: the optomotor response (OMR-Perf = -0.93), indicating that MK-801 causes a generalized decrease in visual responses, perhaps by acting on circuits within the retina.Therefore, based on these experiments with global drug applications, we cannot determine if MK-801 influences the plasticity process in dark-flash habituation, and this is why we did not pursue it further in this project.

Anyway, I hope that you take these suggestions as constructive and, in the spirit that they are intended, as possible routes for improving an already very interesting manuscript.

We are very grateful for your suggestions, which we feel has helped us to improve our manuscript substantially.

**Reviewer #2 (Recommendations For The Authors):**
Overall, the manuscript is well-written, but confusing at times. The results are not always presented in a consistent way, and I found myself having to dig in the raw data or code to find answers. There is a certain disconnect between the free-swimming results, and the calcium imaging, which is somewhat inevitable based on other published work. But I am unsure of what they each bring to the other, as the results from Fig.6 do not match at all the changes observed in the behavioural assays, it almost feels like two separate studies and the inconsistencies make the model appear unlikely.

We agree that there is a disconnect at the behavioural level in our free-swimming and head-embedded imaging experiments. However, this does not necessarily mean that the activity we observe during the imaging experiments cannot be informative about processes that are also occurring in freely-swimming fish. For example, it is possible that the dark-flash circuit is responding and habitating similarly in the head-embedded and freely-swimming preparations, but that in the latter context there is an additional blockade on motor output that massively decreases the propensity of the fish to initiate any movements. In such a case, the “disconnect between the free-swimming results, and the calcium imaging” would indicate that the relationship between neural activity and habituation behaviour is rather complex.

Without a method to record activity from freely swimming fish at our disposal, we can not determine this, one way or the other.

We hope that we now acknowledge these concerns appropriately in the discussion:

“We also found that the same pharmacological treatments that result in strong alterations to habituation behaviour in freely swimming larvae ([fig:5]), resulted in relatively subtle and complex functional alterations in the circuit ([fig:6]). Making direct comparisons between freely-swimming behaviour and head-fixed Ca2+ imaging is always challenging due to the differences in behaviour observed in the two contexts, and therefore our failure to identify a clear logic in these experiments may have technical explanations that will require approaches to measure neural activity from unrestrained and freely-behaving animals to resolve . Alternatively, these results are again consistent with the idea that habituation is a multidimensional and perhaps highly non-linear phenomenon in the circuit, which cannot be captured by a simple model. “

I am not convinced by the results surrounding GABA, from the inconsistent GABA receptor antagonist profile to the post hoc identification of GABAergic neurons as it is currently done in the manuscript. I think that the current focus on GABA does a disservice to the manuscript. However, the novel findings surrounding the potential role of Melatonin, and Estrogen, in habituation are quite interesting.

We agree that we focused too heavily on our hypothesized role for GABA in our original manuscript, and we hope that the reviewer agrees that our updated manuscript is an improvement. We also thank the reviewer for their interest in our Melatonin and Estrogen results, for which follow up studies are ongoing to characterize the effects of these hormones and their receptors on habituation.

There is an assumption that all the adaptation profiles are related to the DF (although that is somewhat alleviated in the discussions of the ON responses) and not to the luminosity changes. But there is no easy way to deconvolve those two in the current experiments. I would like the timing of the fluorescence rise to be quantified compared to the dark flash stimulus onset, potentially spike inference methods could help with giving a better idea of the timing of those responses. Based on the behavioural responses that were <500ms in Randlet O et al, eLife, 2019; we would expect only the fastest DF responses to be linked to the behaviour.

We agree that we are unable to disambiguate responses to the dark flash that initiate the O-bend response, and those that are related to only changes in luminosity. As discussed above, our Ca2+ imaging approach is severely limited in temporal resolution and therefore spike inference methods are not appropriate.

**Major comments**
Fig.1: There seems to be a very variable lag between the motor events and DF responses, furthermore, it does not seem that the motor responses follow a similar habituation rate as in 1Bi. Although this only shows the smoothed 'movement cluster' from the rastermap, it could hide individual variability. It would be important to know what the 'escape' rate was in the embedded experiment, asFig.1 sup.1 seems to indicate there was little to no habituation. It would also be needed to know which motor events are considered linked to the DF stimulus, and how that was decided. Was there a movement intensity threshold and lag limit in the response?

We interpret this concern as relating to the data presented in Figure 6A, where we quantify the habituation rate in the head-embedded experiments. As we have discussed, both above and in the manuscript, we saw very strongly muted responses to DFs in the head-embedded preparation, but we neglected to describe our method of quantifying the responses. We have added the following description to the methods:

“To quantify responses to the dark flash stimuli we used motion artifacts in the imaging data to identify frames associated with movements ([fig:1]-[fig:S1]). Motion artifact was quantified using the “corrXY” parameter from suite2p, which reflects the peak of phase correlation comparing each acquired frame and reference image used for motion correction. The “motion power” was quantified as the standard deviation of a 3-frame rolling window, which was smoothed in time using a Savitzky-Golay filter (window length = 15 frames, polyorder = 2). A response to a dark flash was defined as a “motion power” signal greater than 3 (z-score) occurring within 10-seconds of the dark-flash onset, and was used to quantify habituation in the head-embedded preparation ([fig:6]A).“

Line 94: This seems to be a strong claim based on the sparse presence of non-habituating, or potentiating, neurons in downstream regions. However, these neurons appear to be extremely rare, and as mentioned in my comment above, the behavioural habituation appears minimal. These neurons could encode the luminosity and be part of other responses, such as light-seeking in Karpenko S et al, eLife, 2020 or escape directionality in Heap et al, Neuron, 2018. Furthermore, dimming information has been shown to have parallel processing pathways in Robles E et al, JCN, 2020; so it would make sense that not all the observed responses in this manuscript would be involved in behavioural habituation to dark flashes.

We agree that without functional interventions, we do not know which of the neurons we have categorized are specifically involved in the dark flash response habituation. It is possible that thenon-adapting and potentiating neurons are involved in other behaviours. We have therefore removed this statement.

Line 103: It appears that several of those responses are to the changes in luminosity and not the DF itself, especially the ON and sustained responses. Based on the previous DF habituation study from Randlet O et al, eLife, 2019; the latency of the response is below 0.5s. So the behaviour-relevant responses must only include the shortest latency one, as discussed above.

We appreciate the point that the reviewer is making here, but we are less clear about what the difference between “changes in luminosity” and a “dark flash” response are, since a dark flash consists of a change in luminosity. We take it that the reviewer means the difference between a luminance stimulus that elicits an O-bend, from one that does not. In order to disambiguate the two, one would likely need to use stimuli where the luminosity changes, but do not elicit O-bends.

Perhaps due to the limited temporal resolution of our Ca2+ imaging data, we do not see a clear difference in the onset of the stimulus response for any of the functional clusters that would help us to determine which neurons are more relevant to the acute DF response.

Fig.2B. It is very difficult to make out the actual average z-scored fluorescence, a supplementary figure would help by making these bigger. A plot to quantify the maximum response would also be useful to judge how it changes between the first few and few last DF. Another plot to give the time between the onset of the responses and the onset of the DF stimulus is also needed to judge which cluster may be relevant to the DF escapes observed in the free-swimming experiments.

We agree with the reviewer that interpreting these datasets are challenging. We did include the actual average z-scored fluorescence in Figure 6—figure supplement 1, panel D. This figure also includes a comparison between the predicted Ca2+ response to the dark flash (the stimulus convolved with the approximate GCaMP response kernel), which shows that all OFF-responding neuronal classes show very similar rise time response kinetics, and thus this analysis does not help to judge whether a cluster is more or less relevant to O-bend responses in the free-swimming experiments. We appreciate that there are differences in opinion about the best way to present the data, but we have opted to leave our original presentation.

Line 130: Is a correlation below 0.1 meaningful or significant? It does not seem like this cluster would be a motor or decision cluster.

Our goal with this correlational analysis to motor signals was to identify if certain clusters of DF responsive neurons were more associated with motor output, and therefore may be more downstream in the sensori-motor cascade. Cluster 4 showed the highest median correlation across the population of cells. Whether a median correlation of ~0.1 is “meaningful” is impossible for us to answer, but it is highly “significant” in the statistical sense, as is evident by the 99.99999% confidence intervals plotted. We note that these cells were not selected based on their correlation to the motor stimulus, but only to the dark flash stimulus. There are “motor” clusters that show much higher correlations to the motors signals, as is evident in Figure 1G.

Line 165: Did the changes observed for Pimozide fall below the significance threshold, were lethal, or were the results not repeated? It does not appear in source data 2.

Pimozide was lethal in our screen and therefore does not appear in the source data file. Indeed, in our previous experiments with Pimozide we had already established that a 10uM dose is lethal, and that the maximal effective dose we tried was 1uM as reported in (Randlett et al., Current Biology, 2019).

We have clarified this in the text:

“While the false negative rate is difficult to determine since so little is known about the pharmacology of the system, we note that of the three small molecules we previously established to alter dark flash habituation that were included in the screen, Clozapine, Haloperidol and Pimozide , the first two were identified among our hits while Pimozide was lethal at the 10\muM screening concentration.”

Fig.1B and Fig.3B are the same data, which is awkward and should be explicitly stated. But the legends do not match in terms of the rest period. Which is correct? It is also important to note the other behavioural assays in the 'rest' period.

We thank the reviewer for pointing out this discrepancy in the legend. We have corrected the typo in the figure legend of Figure 3B :

“Habituation results in a progressive decrease in responsiveness to dark flashes repeated at 1-minute intervals, delivered in 4 training blocks of 60 stimuli, separated by 1hr of rest (from 0:00-7:00).”

We have also added a statement that the data is the same as that in Figure 1B.

Figure 3-4: SSMD fingerprint, there is no description of the different behavioural parameters. What they represent is left to the reader's inference. There is no mention of SpontDisp in the GitHub for example, so it is hard to know how these different parameters were measured. Even referring to the previous manuscript on habituation (Randlet O et al, eLife, 2019) does not shed light on most of them, for example, I suppose TwoMvmt represents the 'double responses' from the previous manuscript. Furthermore, there are inconsistencies between 3C and 4B, some minor (SpontDisp becomes SpntDisp), but Curve-Tap has disappeared for example, and I suspect became BendAmp-Tap. A more thorough description of these measures, and making the naming scheme consistent, are essential for readers to know what they are looking at.

We again thank the reviewer for their careful assessment of our data, and we apologize for this sloppiness. We have gone through and made the naming of these parameters consistent in both figures, and have added another supplementary table that describes in more detail what each parameter is, and how it relates to the analysis code (Figure3_sourcedata3_SSMDFingerprintParameters.xls). This was an essential missing piece of information from our original manuscript.

Line 206: While this prioritization makes sense, how was it implemented, how was the threshold decided and which were they? A table, or supplementary figure, would help to clarify the reason behind the choices. Fig.4C being cropped only around the response probability makes it impossible to judge if the criteria were respected, as the main heatmap is too small. For example, the choice of GABA receptor antagonists is somewhat puzzling, as besides PTX it does not seem that the other compounds had strong effects, with Amoxapine for example having seemingly as much effect on Naive and Train, with little in Test. And Bicuculline gave negative SSMD for prob in the three cases.The dose-response for PTX does lend credence to its effect, but I would have liked the other compounds, especially bicuculline. The melatonin results, for example, are much more convincing and interesting in our opinion.

While in hindsight it may have been possible to do the hit prioritization in a systematic way using thresholding and ranking, we did this manually by inspecting the clustered fingerprints. We have clarified this in the text: “This manual prioritization led to the identification of the GABAA/C Receptor antagonists…”

While we agree that it is not possible to judge how well we performed this prioritization based on the images presented, we note that we do provide the full fingerprint data in the supplementary data, for which the reader is welcome to draw their own conclusions.

We have not performed further experiments with amoxapine, so we can not comment further on this. We did perform additional experiments with bicuculline, for which we did see effects similar to those of PTX, were habituation was inhibited. However, the effects are weaker and more variable than what we observe with PTX, and bicuculline also inhibits the initial responses of the larvae, causing their Naive response to be lower. Therefore we did not include it in our manuscript. We include these data here in Author response image 1 to reassure the Reviewer that picrotoxinin is not the only GABA Receptor antagonist for which we see inhibitory effects on habituation.

**Author response image 1. sa4fig1:** 

Fig.6: Why was the melatonin concentration used only 1um instead of 10um on the screen?

Based on dose response experiments (Figure 5B, and others not shown), we found that the effect of Melatonin on habituation saturates at about 1uM, and therefore we used this dose.

Line 277: As the correlation with motor output is marginal at best, and the authors recognize the lack of behaviour in tethered animals, I would be careful about such speculation. Especially since the other changes are complex and go in all directions.

While we appreciate the reviewer's caution, we feel that our statement is appropriately hedged using “might be”. We have also removed the statement “and thus is most closely associated with behavioural initiation”.

We now state:

“However, opposite effects of PTX and Melatonin were observed for 4_L^{strgD} neurons ([fig:6]C), which we found to be most strongly correlated with motor output ([fig:2]F). Therefore, this class might be most critical for habituation of response Probability.”

Fig.7: I am not sure how convincing these results are. 7F may have been more convincing, but to be thorough the authors would need to register the Gad1b identity to the calcium imaging and use their outline to extract the neuron's fluorescence. As it is, in the tectum, it is hard to be sure that all the identified neurons are indeed Gad1b positive, as that population is intermingled with other neuronal populations. The authors should consider the approach of Lovett-Barron M et al, Nat Neuro, 2020. Alternatively, the authors can tone down the language used in this section to match the confidence level of the association they propose.

Figure 7A-E are what can be considered “virtual colocalization” analyses, where we are comparing the localization of data acquired in different experiments using image registration to common atlas coordinates. We agree that these results alone will never be very strong evidence for the identification of individual cells. The MultiMAP approach of Lovett-Barron is a powerful approach, though it makes the assumption that registration accuracy will be subcellular, which in practice may often not be the case. We believe that a better approach is to label the cells of interest during the Ca2+ imaging experiment itself, as we did 7F and G. The challenge in this experiment is binarizing the ROIs and thus deciding what is and is not a Gad1b-positive cell. In our opinion, the fact that these two independent experiments came to the same conclusion regarding Cluster 10 and 11 is good evidence that these cell types are likely predominantly GABAergic.

As discussed above, we have re-written the manuscript to tone down our claims about the role of GABA and GABAergic neurons in habituation, which we hope the reviewer will agree better reflects the limitations of the data in Figure 6 and 7.

Line 317: Based on the somewhat inconsistent results of the other GABA antagonists, I would be careful. Picrotoxin has been reported to antagonize other receptors besides GABA, see Das P et al, Neuropharma, 2003. So the results may be explained by a complex set of effects on multiple pathways with PTX.

Off target effects are an important concern with any pharmacological experiment, and perhaps especially in zebrafish where receptors and targets can be quite divergent from those in mammals where most drug targets have been characterized. We have added this sentiment to the discussion:

“We cannot rule out the possibility that off-targets of PTX, or subtle non-specific changes in excitatory/inhibitory balance alter habituation behaviour.”

Line 400-403, 430: There are some conflicting statements regarding the potential role of clusters 1 and 2 in DF habituation. Do the authors think they play a role in the behaviour measured in this manuscript? Could they clarify what they mean?

We see how our original statement in line 429 about the presence of cluster 1 and 2 neurons in the TL implied a role in dark flash habituation. This was not our intent, and we have removed “which also contains high concentrations of on-responding 1OnnoA,2OnmedD neurons”.

Our thoughts on these neurons are now stated in the discussion as:

“We also observed classes exhibiting an On-response profile (1OnnoA and 2OnmedD ). These neurons fire at the ramping increase in luminance after the DF, making it unlikely that they play a role in aspects of acute DF behaviour we measured here. These neurons exist in both non-adapting and depressing forms suggesting a yet unidentified role in behavioural adaptation to repeated DFs.“

Minor commentsLine 73 (and elsewhere): Why use adaptation instead of habituation (also in the adaptation profile)? Do you suspect your observations do not reflect habituation, but a sensory adaptation mechanism?

We have used the convention that “habituation” refers to observations at the behavioural level, while “depression” and “potentiation” refer to observations at the neuronal level. We use the term “adaptation” to refer to neuronal adaptations of either sign (depression or potentiation), as in line 73.

We believe that our observations reflect neuronal adaptations that underlie habituation behaviour.

Line 71: It is debatable that the strongest learning happens in the first block, the difference between the first and last response seems to grow larger with each successive block. What do the authors mean by 'strongest'

We agree that “strongest” was ambiguous. We have changed this to “initial”:

“We focused on a single training block of 60 DFs to identify neuronal adaptations that occur during the initial phase of learning ”

Fig.1F: there is no rastermap call in the GitHub repository, was the embedding done in the GUI? If so, it should also be shared for reproducibility's sake.

Yes, Fig.1F was created using the suite2p GUI, as we have now clarified in the methods:

“The clustered heatmap image of neural activity ([fig:3]F) was generated using the suite2p GUI using the “Visualize selected cells” function, and sorting the neurons using the rastermap algorithm ”

The image is available in the “Figure1 - Ca2Imaging.svg” file available here: https://github.com/owenrandlett/lamire_2022/tree/main/LamireEtAl_2022

Line 101: while true that AffinityPropagation does not require input on the number of clusters, preference can influence the number of clusters. It seems that at least two values were tested in the search for the clusters, can the authors comment on how many clusters the other preference value converged (or failed to converge) on?

Indeed, as with any clustering approach, the resultant clusters are highly dependent on the input parameters, in this case the “preference”, as well as “damping” and the choice of affinity metric. By varying these parameters one can arrive at anywhere between 2 and hundreds of clusters.

It is for this reason that we feel that the anatomical analyses of these clusters is very important, making the assumption that neurons of differing functional types will have different localizations in the brain, as we explained in the Results:

“While these results indicate the presence of a dozen functionally distinct neuron types, such clustering analyses will force categories upon the data irrespective of if such categories actually exist. To determine if our cluster analyses identified genuine neuron types, we analyzed their anatomical localization ([fig:2]C-E). Since our clustering was based purely on functional responses, we reasoned that anatomical segregation of these clusters would be consistent with the presence of truly distinct types of neurons.”

We also acknowledge in the Results that the clustering approach has limitations:

“These results highlight a diversity of functional neuronal classes active during DF habituation. Whether there are indeed 12 classes of neurons, or if this is an over- or under-estimate, awaits a full molecular characterization. Independent of the precise number of neuronal classes, we proceed under the hypothesis that these clusters define neurons that play distinct roles in the DF response and/or its modulation during habituation learning“

Fig.2. My understanding is that the cluster numbers are arbitrary unless there is a meaning to them, which then should be explained. I would recommend grouping the clusters per functional category as in Fig.6 to make it easier for the reader.

Cluster number reflects the ordering in the hierarchical clustering tree shown in Figure 2B. We feel that this is the most logical representation of their functional similarity. We have clarified this in the Methods:

“ We then used the Affinity Propagation clustering from scikit-learn , with “affinity” computed as the Pearson product-moment correlation coefficients (corrcoef in NumPy ), preference=-9, and damping=0.9, and clustered using Hierarchical clustering (cluster.hierarchy in SciPy ). Cluster number was assigned based on the ordering of the hierarchical clustering tree. ”

Fig.3 SSMD fingerprint, it would be much easier for the readers if the list of parameters was clearer and rotated 90 degrees. Maybe in a supplementary figure to show what each represents.

We agree that the SSMD fingerprint is very difficult to interpret. As discussed above, we have now included a supplementary table (Figure3_sourcedata2_SSMDFingerprintParameters.xlsx) where we have clarified what each parameter represents.

Fig.4: The use of the same colours across the clustering methods is confusing, especially after the use of colours for the SSMD fingerprint in Fig.3. and at the bottom of 4A. Fig.4A for example could have been colour coded according to the most affected behaviour in the fingerprint at the bottom.Fig.4B the coloured text is difficult to read, especially for the lighter colours.

We agree that our use of color is not perfect, but we have attempted to use them consistently: for example when referring to a functional cluster, or a drug manipulation. We don’t think that there is a sufficient number of distinguishable colors for us to never use the same color twice.

Fig.4C if the goal is to show similarity, the relevant drugs could be placed adjacent to each other. One could also report the Euclidean distance, or compute how correlated the different fingerprints are within one pharmacological target space.

The goal of Fig 4C is to highlight where Bicuculline, Amoxapine, Picrotoxinin, Melatonin, Ethinyl Estradiol and Hexestrol lie within the clustered heatmap of the behavioural fingerprints (Fig 4A), and

demonstrate how the probability of response to dark flashes is modulated by these drugs. In our analyses, “similarity” is a function of the clustering distance.

Fig.6D 'Same data as M, ...' I assume should be 'Same data as C,...'

Indeed, thank you for pointing out this error that we have corrected.

Fig. 7 How many GCaMP6s double transgenic larvae were imaged?

6 fish were imaged, as is stated in the legend to Fig 7G

Line 407: all is repeated.

We apologize, but we do not see what is repeated at line 407. Can you please clarify?

Line 481: Would testing spontaneous activity after training for 7h be unbiased, could there be fatigue effects?

We tested for fatigue effects in our previous study, comparing larvae that received the training for 7hrs and those that did not, and we saw no deficits in spontaneous activity, tap response, or OMR performance (Figure S1, Randlett et al., Current Biology, 2019).

Line 610: There are some inconsistencies between the authors' contributions in the manuscript and the one provided to eLife.

Thank you, we will double check this in the resubmission forms. The authors' contributions in the manuscript are correct.

**Reviewer #3 (Recommendations For The Authors):**
I would rather recommend the authors divide this manuscript into two and publish two papers by adding some more strengthening data for each part such as cellular manipulations, e.g. ablation to prove the critical involvement of 12(Pot, M) neurons in habituation.

We thank the reviewer for their suggestion, but have opted not to split the paper into two. We feel that the collective message of this paper and approach combining molecular and functional analysis will be of interest, and we believe the incongruencies in our results reflects the complexity inherent within the system.